# TET enzymes regulate skeletal development through increasing chromatin accessibility of RUNX2 target genes

Lijun Wang [1,5], Xiuling You[2,5], Dengfeng Ruan [3,4,5], Rui Shao[1], Hai-Qiang Dai [2], Weiliang Shen[3], Guo-Liang Xu [2], Wanlu Liu [3,4] ✉ & Weiguo Zou [1,2] ✉

The Ten-eleven translocation (TET) family of dioxygenases mediate cytosine demethylation by catalyzing the oxidation of 5-methylcytosine (5mC). TET-mediated DNA demethylation controls the proper differentiation of embryonic stem cells and TET members display functional redundancy during early gastrulation. However, it is unclear if TET proteins have functional significance in mammalian skeletal development. Here, we report that *Tet* genes deficiency in mesoderm mesenchymal stem cells results in severe defects of bone development. The existence of any single *Tet* gene allele can support early bone formation, suggesting a functional redundancy of TET proteins. Integrative analyses of RNA-seq, Whole Genome Bisulfite Sequencing (WGBS), 5hmC-Seal and Assay for Transposase-Accessible Chromatin (ATAC-seq) demonstrate that TET-mediated demethylation increases the chromatin accessibility of target genes by RUNX2 and facilities RUNX2-regulated transcription. In addition, TET proteins interact with RUNX2 through their catalytic domain to regulate cytosine methylation around RUNX2 binding region. The catalytic domain is indispensable for TET enzymes to regulate RUNX2 transcription activity on its target genes and to regulate bone development. These results demonstrate that TET enzymes function to regulate RUNX2 activity and maintain skeletal homeostasis.

DNA methylation is a prominent epigenetic modification in the mammalian genome. The ten-eleven translocation (TET) family, which includes TET1, TET2, and TET3 in mammals, iteratively oxidizes 5-methylcytosine (5mC) into three different products including 5-hydroxymethylcytosine (5hmC), 5-formylcytosine (5fC) and 5-carboxylcytosine (5cac), which finally lead to DNA demethylation[1–3]. DNA demethylation can regulate gene expression by providing potential open chromatin landscape to transcriptional regulators[4–6]. The binding of transcription factors or RNA polymerase to promoter or enhancer regions contributes to the transcription complex assembly and gene expression regulation. Dynamic changes in DNA methylation landscape are essential in regulating the temporal expression of genes during development. Loss of individual *Tet* or combined deficiency of

[1]Institute of Microsurgery on Extremities, Shanghai Jiao Tong University Affiliated Sixth People's Hospital, Shanghai 200233, China. [2]State Key Laboratory of Cell Biology, Shanghai Institute of Biochemistry and Cell Biology, CAS Center for Excellence in Molecular Cell Science, Chinese Academy of Sciences, University of Chinese Academy of Sciences, 320 Yueyang Road, Shanghai 200031, China. [3]Department of Orthopedic Surgery, the Second Affiliated Hospital, Zhejiang University School of Medicine, Zhejiang 310009, China. [4]Zhejiang University-University of Edinburgh Institute (ZJU-UoE Institute), Zhejiang University School of Medicine, International Campus, Zhejiang University, 718 East Haizhou Road, Haining 314400, China. [5]These authors contributed equally: Lijun Wang, Xiuling You, Dengfeng Ruan. ✉e-mail: wanluliu@intl.zju.edu.cn; zouwg94@sibcb.ac.cn

*Tet1/2* allows for embryogenesis, whereas *Tet1/2/3* triple-knockout (TKO) mouse embryonic stem cells (ESCs) deplete 5hmC and impair ESC differentiation[7–9]. Consistently, germline-specific conditional knockout of *Tet1/2/3* results in primitive streak patterning defects associated with impaired maturation of axial mesoderm and failed specification of paraxial mesoderm, which depends on TET catalytic domain[10]. However, it remains to be determined whether TET proteins are functionally redundant in regulating skeletal development.

During mammalian embryogenesis, two types of bone formation have been identified[11]. Long bones are formed initially as a framework, or anlage of hyaline cartilage originated from mesenchymal stem cells (MSCs) and then replaced by mineral deposition of osteoblasts in a process known as endochondral ossification[12]. Flat bones are produced by intramembranous ossification, which involves the direct development of osteoblasts from mesenchymal stem cells without an intervening cartilage model[12]. In both forms of bone formation, MSCs are of great importance. MSCs are multipotent progenitors that can differentiate into several distinct cell types, including osteoblasts, chondrocytes, and adipocytes[13]. The lineage commitment mainly depends on specific transcription factors, including RUNX2, SOX9, PPARγ, etc.[14–16]. RUNX2 is one of the key transcription factors for osteoblast differentiation and bone development[16–18]. The skeletal mineralization of heterozygous *Runx2* knockout mice is impaired, with delayed closure of the fontanels and clavicular hypoplasia, resembling the cleidocranial dysplasia (CCD) phenotype[19]. A previous study indicated that RUNX2 mutations were detected in 70% of patients with the CCD. Whereas no RUNX2 mutations have been identified in the rest 30% of patients[20]. The mechanism in these individuals remains elusive, suggesting that other biological processes are involved in this disease. A recent study suggested that DNA hypermethylation in *Tet1/2* double knockout mice leads to osteoporosis by inhibiting the expression of *Runx2*[21], indicating the potential critical roles of TET proteins in skeletal development.

In this study, we show that conditional knockout of *Tet1/2/3* in mesenchymal stem cells with *Prx1-Cre* leads to severe defects in bone development, including delayed closure of the fontanels, clavicular hypoplasia, and shortened limbs. Importantly, the retention of any allele of the *Tet* genes can maintain bone formation, suggesting functional redundancy of TET proteins in bone development. To uncover the potential mechanism of TET proteins regulating bone formation, we performed multi-level genomics experiments and combinatory analysis with WGBS, 5hmC-Seal, ATAC-seq, ChIP-seq, and RNA-seq. Our results suggest that RUNX2 may recruit TET enzymes to modulate the epigenetic landscape of ossification-related genes, thus regulating bone formation.

## Results

### Hypomethylation facilitates osteoblast differentiation

To reveal the function of the dynamic of DNA methylation on osteogenesis, we treated primary mouse osteoblasts with 5-Azacytidine (5-Aza), which caused DNA demethylation or hemi-demethylation in the osteoblasts (Supplementary Fig. 1a). The treatment of 5-Aza could promote osteoblast differentiation, as demonstrated by the increased ALP activity at day 7 (Supplementary Fig. 1b, top and c) and Alizarin red S signals at day 21 (Supplementary Fig. 1b, bottom). The expression of a series of osteogenic marker genes, including Sp7 transcription factor (*Sp7*), alkaline phosphatase (*Alpl*), Collagen Type I Alpha 1 Chain (*Col1α1*), and Osteocalcin (*Bglap*) (Supplementary Fig. 1d) were increased in 5-Aza treated osteoblasts, suggesting DNA hypomethylation would promote osteoblast differentiation in vitro.

### Loss of TET enzymes impairs bone formation

To further characterize the function of DNA methylation in bone formation in vivo, we utilized *Tet1*, *Tet2,* and *Tet3* triple conditional knockout (TCKO) mice, obtained by breeding *Prx1-Cre* mice with *Tet1*^fl/fl *Tet2*^fl/fl *Tet3*^fl/fl mice[22]. The tissue-specific knockout efficiency was tested by RT-qPCR of *Tet1*, *Tet2*, and *Tet3* in calvarial bones, femurs, and livers (Supplementary Fig. 2a, b). The residual expression of *Tet1*, *Tet2*, and *Tet3* in the femurs of the TCKO mice may be from the presence of other cell types in the femurs. The TCKO mice were extremely short-limbed and weak at 3-week (Supplementary Fig. 2c). TCKO mice were stunted, while retention of any one of *Tet* alleles maintains a normal appearance in mice at postnatal day 0 (P0) (Fig. 1a). To observe the mouse skeletons, we performed whole mount staining by Alcian blue-Alizarin red S staining with P0 mice. Our staining demonstrated that TCKO mice exhibited shortened clavicles (Fig. 1b, c) and long bones including femur (Fig. 1b, d), tibia (Fig. 1b, e), radius (Fig. 1b, f), and humerus (Fig. 1b, g). Also, the closure of the fontanel was delayed (Fig. 1b, h, i). Except for TCKO, *Prx1-cre, Tet1*^fl/+ *Tet2*^fl/fl *Tet3*^fl/fl mice also showed some slight defects in radius (Fig. 1b, f) and humerus (Fig. 1b, g). The closure of fontanel of *Prx1-Cre, Tet1*^fl/+ *Tet2*^fl/fl *Tet3*^fl/fl mice was also slightly delayed at P0, but was compensated at P10 (Fig. 1h). Based on these observations, we conclude that there is a potential functional redundancy of the three TET proteins in mouse bone development.

Loss of all three *Tet* genes in the bone gave rise to the increased 5mC and decreased 5hmC, compromising the DNA methylation dynamics in both the limb and the calvarial bone (Fig. 2a, b). To determine the function of TET proteins in intramembranous and endochondral bone formation, we did a series of histology analyses in calvarial bone and limb. Mineral deposition (Fig. 2c), Osteopontin (OPN), and BGLAP expression (Fig. 2d and Supplementary Fig. 2d) were decreased in the calvarial bone of TCKO mice. To investigate the function of TET proteins during osteoblast differentiation, we isolated primary osteoblasts from the calvarial bone of both WT and TCKO mice. The differential potential was severely disrupted in *Tet*-deficient cells, determined by the significantly decreased ALP activity (Fig. 2e top and f) and calcium nodule formation (Fig. 2e bottom). The declined osteogenesis in TCKO was also demonstrated by the strongly decreased expression of a series of osteogenic marker genes, including *Runx2*, *Sp7*, *Col1α1*, *Alpl,* and *Bglap* (Fig. 2g). These data demonstrate that the intramembranous bone formation was impaired.

Additionally, we found that the length of the femur was shorter in TCKO mice from E15.5 to P0 compared to the WT mice (Supplementary Fig. 3a). Then the endochondral ossification process was investigated. At E13.5, the chondrocyte in the center of the femur of WT mice was under hypertrophic process whereas cells in TCKO mice resembled undifferentiated mesenchymal cells (Supplementary Fig. 3a). The primary ossification center became mature in WT mice but remained occupied by chondrocytes in TCKO mice at E16.5 (Supplementary Fig. 3a–c). Consistently, more Col10a1 and Mmp13 signals were present in the bone marrow of TCKO mice compared with WT mice at E16.5 and E17.5 (Supplementary Fig. 3b–d). In vitro, the chondrocyte differentiation was also impaired in TCKO cells, compared with wild-type chondrocytes (Supplementary Fig. 3k, l). These data proved that chondrocyte differentiation and maturation were impaired by the loss of TET proteins. Subsequently, we checked whether the bone formation process was also delayed on account of impaired chondrocytes in TCKO mice. As expected, TCKO mice exhibited less mineralization in midshaft of the femur at E16.5 (Fig. 2h, Supplementary Fig. 3a). The expression of BGLAP and OSX was also decreased in the femurs of TCKO mice (Supplementary Fig. 3e–g). We also observed increased PCNA-positive cells and cleaved Cas3-positive cells in the chondrocyte zone of TCKO mice (Supplementary Fig. 3h–j). The impaired chondrocyte maturation and bone formation may give rise to the shortened limbs of TCKO mice. The osteoclasts were not significantly affected at P0 (Supplementary Fig. 4a, b). In conclusion, TET proteins are essential for both intramembranous and endochondral bone formation.

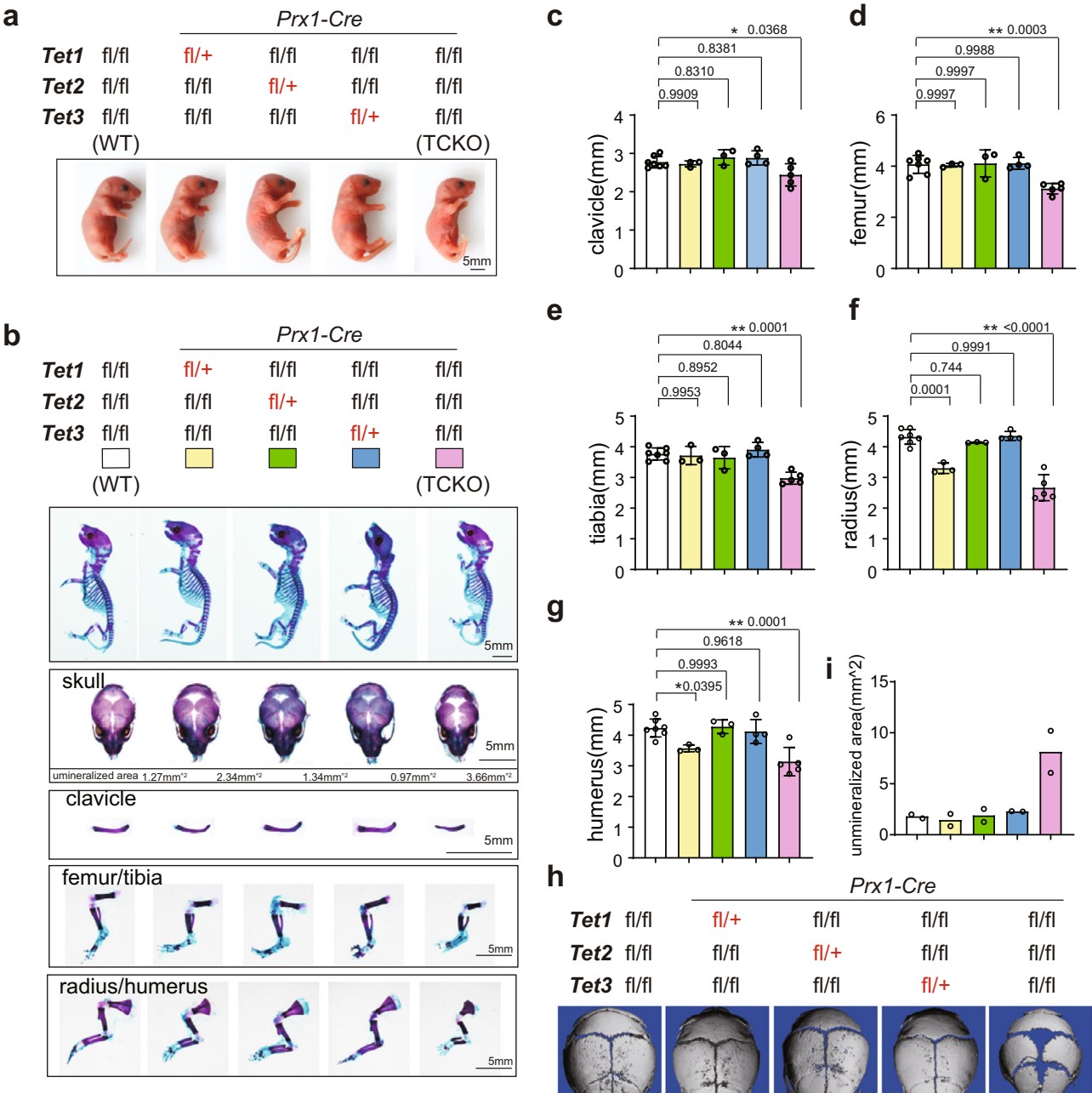

**Fig. 1 | Loss of *Tet* family genes leads to cleidocranial dysplasia-like phenotype.**
**a** Gross images of indicated genotype mice. Scale bar = 5 mm. **b** The whole mount skeleton staining of WT and indicated genotype mice by Alcian blue and Alizarin red S at postnatal day 0 (P0). The quantification data of the unmineralized area was annotated below the cranial bones. scale bar = 5 mm. **c–g** Quantification of the length of clavicle (**c**), femur (**d**), tibia (**e**), radius (**f**), and humerus (**g**) of indicated genotype mice. *$P < 0.05$, **$P < 0.01$. Ordinary one-way ANOVA. Data are presented

as mean ± s.d., WT mice = 7, TCKO mice = 5, *Prx1-Cre; Tet1$^{fl/+}$ Tet2$^{fl/fl}$ Tet3$^{fl/fl}$* mice=3, *Prx1-Cre; Tet1$^{fl/fl}$ Tet2$^{fl/+}$ Tet3$^{fl/fl}$* mice = 3, *Prx1-Cre; Tet1$^{fl/fl}$ Tet2$^{fl/fl}$ Tet3$^{fl/+}$* mice = 4 biologically independent animals. **h** 3D μ-CT images of cranial bones isolated from indicated genotype mice at postnatal day 10 (P10). Representative images are from two independent samples. **i** Quantification of the unmineralized area of cranial bones in **h**. $n$ = 2 biologically independent animals.

## Impaired bone formation by *Tets* deficiency is dependent on catalytic domain

TET enzymes are multiple domain proteins and could function both dependent and independent of their dioxygenase activity[10,23]. To further examine whether impaired bone formation by TET is dependent on its demethylase activity, we overexpressed the catalytic domain (CD) and its enzyme inactivated mutation (HD) of TET1, TET2, and TET3[24,25] to rescue the defects of osteoblast differentiation in *Tet*-deficient cells. We observed that only CDs, but not HDs could partially restore the defects of osteogenesis, assessed by increased ALP activity (Fig. 3a top and b) and calcium nodule formation (Fig. 3a

bottom). Among the three CD domains, TET2-CD had the most impact than Tet1-CD and Tet3-CD on calcium nodule formation in the late stage of osteogenic differentiation (Fig. 3a bottom). To further confirm the dependence of dioxygenase activity for TET proteins on bone development in vivo, we selected the *Prx1-Cre, Tet1$^{fl/HD}$ Tet2$^{fl/fl}$ Tet3$^{fl/fl}$* mice as an in vivo model, in which one of *Tet1* allele was replaced by an enzyme inactivated mutation and others were conditional knocked out. As shown in Fig. 3c–h, *Prx1-Cre, Tet1$^{fl/HD}$ Tet2$^{fl/fl}$ Tet3$^{fl/fl}$* mice, and TCKO mice showed almost equivalent shortened clavicles and limbs and delayed closure of the fontanels. Furthermore, osteoblasts isolated from *Prx1-Cre, Tet1$^{fl/HD}$ Tet2$^{fl/fl}$ Tet3$^{fl/fl}$* mice showed impaired

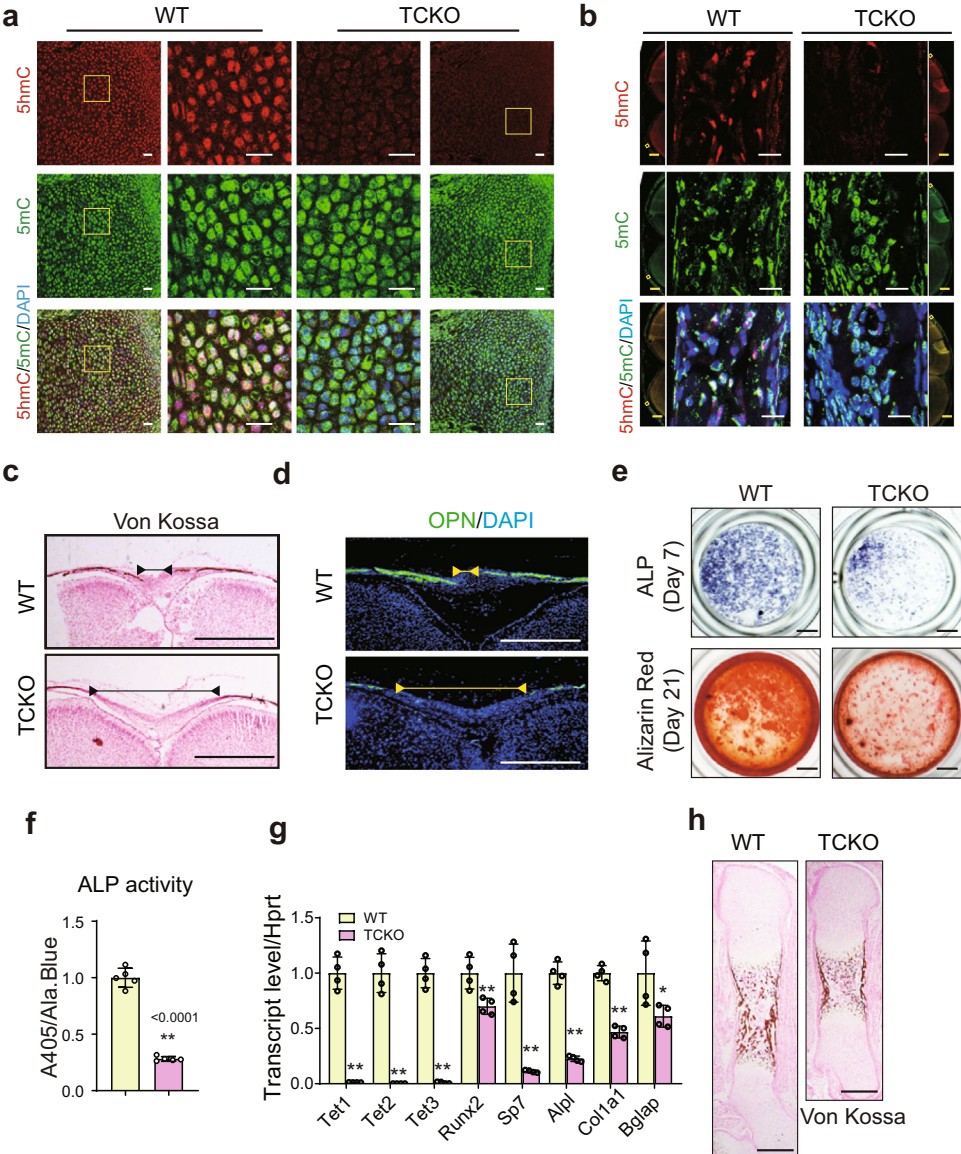

**Fig. 2 | Loss of *Tet* family genes impairs bone formation.**
**a**, **b** Immunofluorescence assay of 5mC and 5hmC in limb at embryonic day 16.5 (E16.5) (**a**) and calvarial bone at P0 (**b**) of WT and TCKO mice. Scale bar = 20 μm or 500 μm. Representative images for three independent samples. The images in the center were zoomed in from the flanking images. **c** Representative images of Von-kossa staining in the cranial bones from WT and TCKO mice at P0. Scale bar = 500 μm. **d** Immunofluorescence of bone formation marker Osteopontin (OPN) in the cranial bones from WT and TCKO mice at P0. Scale bar = 500 μm. The lines with endpoint indicated the unmineralized region in the calvarial bone. Representative images for three independent samples. **e** ALP staining and Alizarin red S staining

after osteoblast differentiation for 7 days (top) and 21 days (bottom), respectively. Scale bar = 1 mm. Representative images for five independent samples. **f** ALP activity quantification was measured by phosphatase substrate assay as A405/Ala.Blue. **P < 0.01. Two-tailed Student's *t* test. Data are presented as mean ± s.d., *n* = 5 independent cell supernatants. **g** RT-qPCR analysis of *Tet1*, *Tet2*, *Tet3*, *Runx2*, *Sp7*, *Alpl*, *Col1α1*, and *Bglap* expression after osteoblast differentiation for 7 days, cells were from WT and TCKO mice. *P < 0.05, **P <0.01. Two-tailed Student's *t* test. Data are presented as mean ± s.d., *n* = 4 independent samples. **h** Representative images of Von-kossa staining in the femurs of WT and TCKO mice at E16.5. Scale bar = 500 μm.

osteogenic ability, determined by decreased ALP activity (Fig. 3i top and j) and calcium nodule formation (Fig. 3i bottom) compared with osteoblasts isolated from WT mice. The expression of a series of osteogenic marker genes, including *Sp7*, *Col1α1*, *Alpl*, and *Bglap* (Fig. 3k) confirmed the osteogenic defects of *Prx1-Cre, Tet1*^fl/HD^ *Tet2*^fl/fl^ *Tet3*^fl/fl^ mice. The osteogenic defects of osteoblasts isolated from *Prx1-Cre, Tet1*^fl/HD^ *Tet2*^fl/fl^ *Tet3*^fl/fl^ mice were more severe than that from TCKO mice (Fig. 3i–k). More profound effects on DNA methylation status caused by the early inactivation of TET1 from germ cells could partially explain the difference between *Prx1-Cre, Tet1*^fl/HD^ *Tet2*^fl/fl^ *Tet3*^fl/fl^ mice and TCKO mice. In conclusion, the dioxygenase activity of TET is required to regulate bone development.

## TET proteins regulate osteogenic genes by opening the chromatin accessibility

To further elucidate how TET proteins regulate the bone formation, we isolated the osteoblasts from WT and TCKO mice to perform RNA sequencing (RNA-seq)[26], whole genome bisulfate sequencing (WGBS)[27], 5hmC-Seal[28] and Assay for Transposase-Accessible Chromatin (ATAC-seq)[29]. Principle component analysis (PCA) showed that the global transcriptome changed dramatically between WT and TCKO cells (Supplementary Fig. 5a), indicating the potential significant function of TET proteins in the primary osteoblasts. We identified 432 downregulated and 431 up-regulated genome-wide significant differential expressed genes (fold change greater than 2 and FDR less than 0.05) in TCKO cells (Fig. 4a). Gene ontology (GO) enrichment analysis

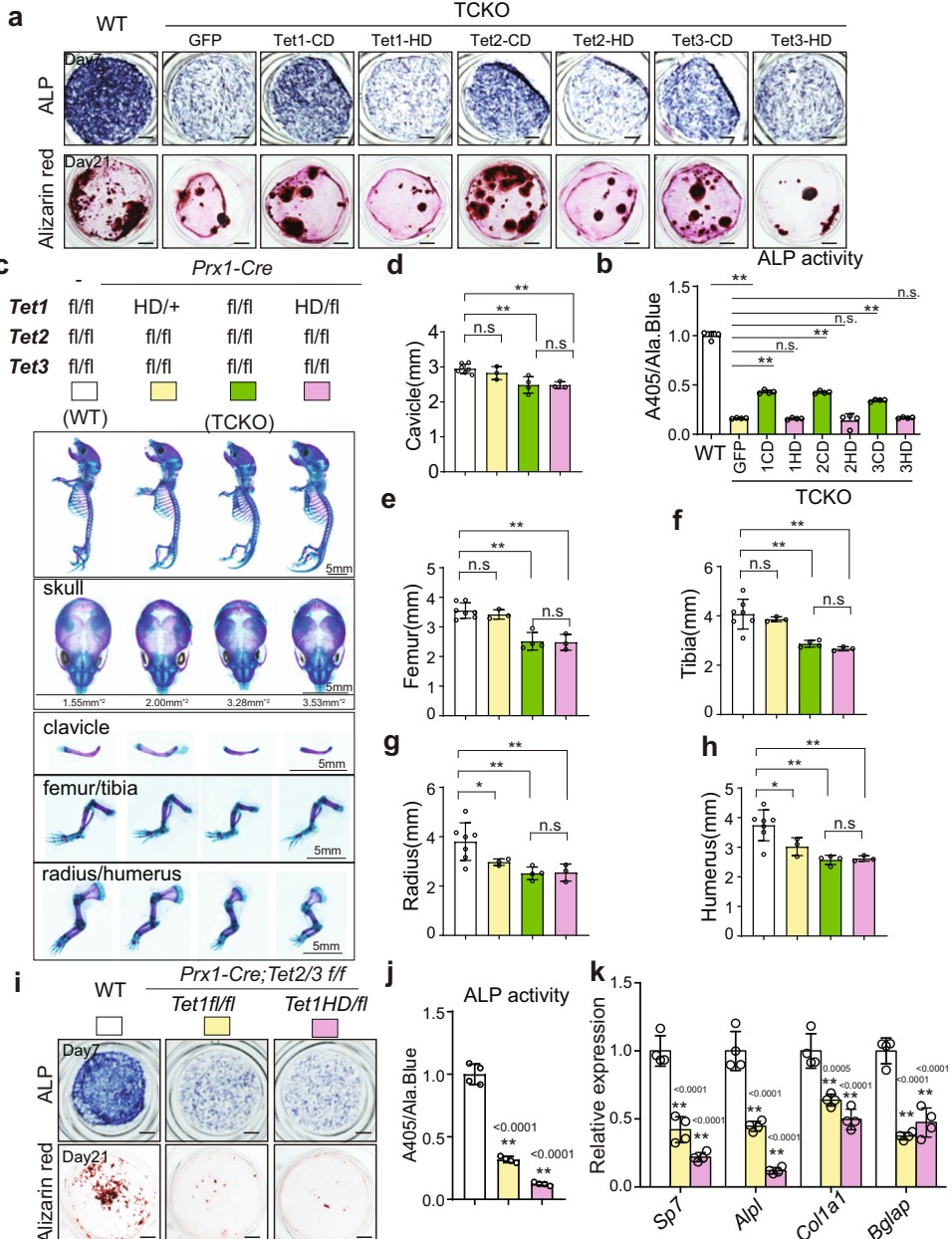

**Fig. 3 | Tet-mediated bone formation is dependent on its catalytic domain.**
**a** ALP staining and Alizarin red S staining after osteoblast differentiation for 7 days (up) and 21 days (bottom), respectively. Cells were isolated from calvarial bones of WT and TCKO mice, and cells from TCKO mice were infected with indicated lentivirus. Scale bar = 1 mm. **b** ALP activity quantification was measured by phosphatase substrate assay as A405/Ala. Blue. *$P < 0.05$, **$P < 0.01$. Ordinary one-way ANOVA. Data are presented as mean ± s.d., $n = 4$ independent cell supernatants. **c** The whole mount skeleton staining of WT and indicated genotype mice by Alcian blue and Alizarin red S at P0. Scale bar = 5 mm. **d–h** Quantification of the length of clavicle (**d**), femur (**e**), tibia (**f**), radius (**g**), and humerus (**h**) of indicated genotype mice. *$P < 0.05$, **$P < 0.01$. Two-tailed Student's $t$ test. Data are presented as

mean ± s.d., WT mice = 7, *Prx1-Cre; Tet1 HD/+ Tet2 f/f Tet3 f/f* mice = 3, *Prx1-Cre; Tet1 f/f Tet2 f/f Tet3 f/f* mice = 4, *Prx1-Cre; Tet1 HD/f Tet2 f/f Tet3 f/f* mice = 3 biologically independent animals. **i** ALP staining and Alizarin red S staining after osteoblast differentiation for 7 days (up) and 21 days (bottom), respectively. Scale bar = 1 mm. **j** ALP activity quantification was measured with by phosphatase substrate assay as A405/Ala. Blue. *$P < 0.05$, **$P < 0.01$. Ordinary one-way ANOVA. Data are presented as mean ± s.d., $n = 4$ independent cell supernatants. **k** RT-qPCR analysis of *Sp7, Alpl, Col1α1,* and *Bglap* expression after osteoblast differentiation for 7 days, cells were from indicated genotype mice. *$P < 0.05$, **$P < 0.01$. Ordinary one-way ANOVA. Data are presented as mean ± s.d., $n = 4$ independent samples.

of these different expression genes (DEGs) indicated that bone morphogenesis and ossification were top enriched GO terms for downregulated genes in TCKO cells, while axonogenesis related GO terms enriched in up-regulated genes (Fig. 4b and Supplementary Fig. 5b). To assess the signaling pathways enriched in those DEGs, we performed KEGG pathway analysis. We identified PI3K–Akt signaling pathway as the most enriching KEGG pathway in downregulated genes in TCKO cells (Fig. 4c), which has previously been reported to associate with the regulation of osteoblast and osteoclast[30,31]. For up-regulated genes,

neuroactive ligand-receptor interaction, PI3K–Akt signaling pathway, and MAPK signaling pathway were the top three KEGG pathways enriched (Supplementary Fig. 5c). Based on WGBS results, we observed a genome-wide 6% hyper CG methylation in TCKO cells (Fig. 4d). The non-CG methylation such as CA and CY methylation displayed little change in TCKO cells (Supplementary Fig. 5d, e). Furthermore, we calculated CG methylation over each chromosome and discovered that all chromosomes except Y chromosome showed a consistent higher CG methylation level in TCKO cells (Supplementary Fig. 5f). The

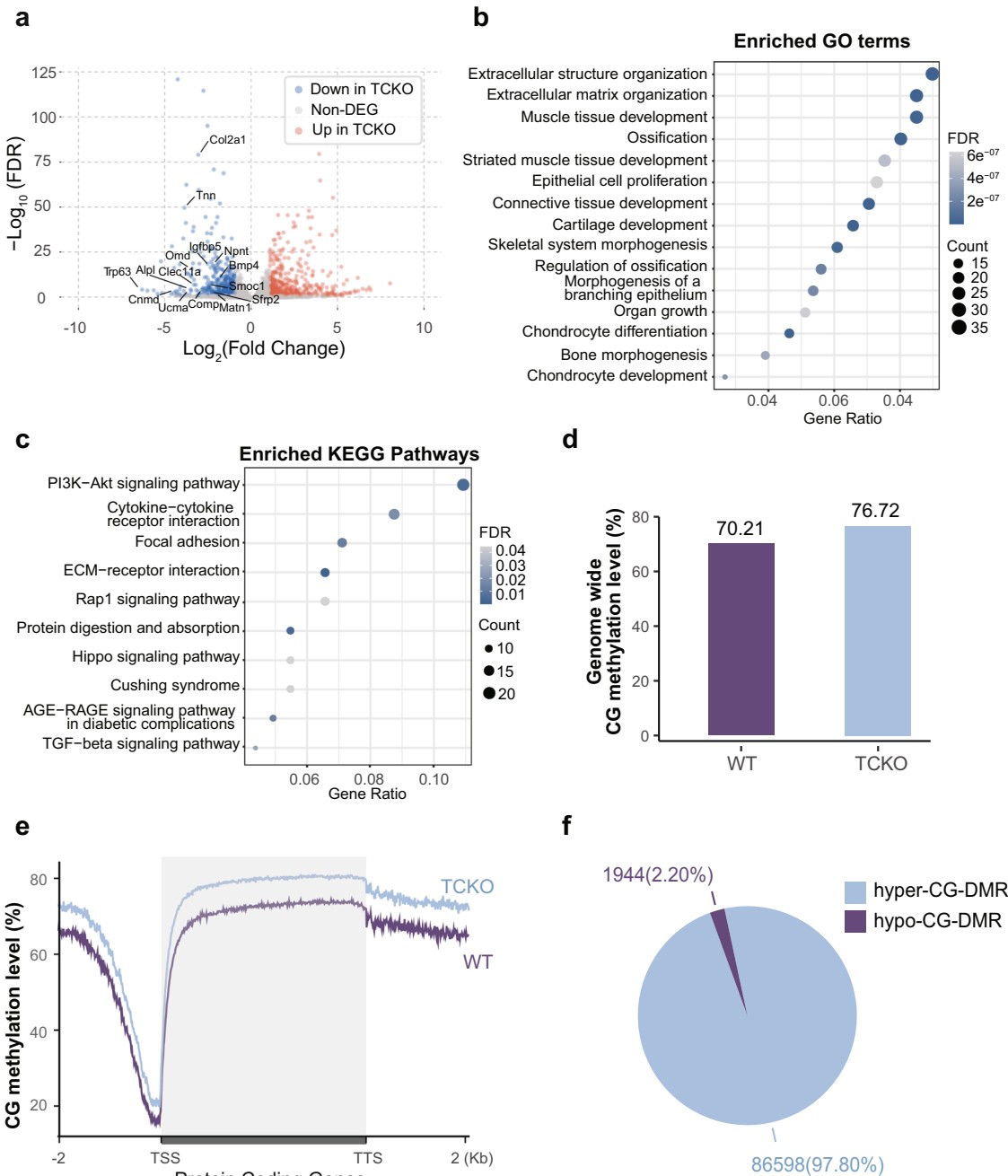

**Fig. 4 | Genome-wide differentially gene expression and DNA methylation patterns between TCKO and WT cells. a** Volcano plot of differential expressed protein-coding genes (Fold Change greater than 2 and FDR less than 0.05). **b**, **c** Enriched GO term (**b**) and KEGG pathway (**c**) analysis of downregulated genes in TCKO cells. **d** Barplot of genome-wide CG methylation level in TCKO and WT cells. **e** Metaplot of CG methylation level over protein-coding genes in TCKO and WT cells. **f** Pie chart of hypermethylated and hypomethylated CG-DMR counts in TCKO and WT cells. TSS transcription start site, TTS transcription termination site.

abnormal pattern we observed here might be due to the fundamentally different composition of murine Y chromosome when compared to other chromosomes[32]. Metaplot of CG methylation over protein-coding genes confirmed a similar pattern in WT and TCKO except for a higher level of CG methylation in both promoters and gene bodies in TCKO cells (Fig. 4e). These results indicated that loss of TET proteins might inhibit the expression of certain genes by increasing promoter CG DNA methylation. Thus, we performed a differential methylated region (DMR) analysis comparing WT and TCKO WGBS data. We tried to define DMR with different thresholds ranging from 20% to 90% CG methylation level difference (Supplementary Fig. 5g). Our analysis indicated that the number of hyper-CG-DMRs overlapped with chromatin Open-to-Close-Regions (oC-Rs) became saturated when we used

0.4 as the methylation difference cutoff. Finally, we defined DMR as at least four CG sites in 200 bp bins, minimal CG methylation level difference >40%, and an FDR <0.05[33]. Genome-wide TCKO cells showed much higher numbers of hyper-CG-DMR (n = 86598) than hypo-CG-DMR (n = 1944) (Fig. 4f). Further analysis showed that the hyper-CG-DMR mainly enriched over genic regions including ncRNA, intron, promoter, and 3'UTR TTS (Supplementary Fig. 5h). To investigate the association of hyper CG methylation with DEGs in TCKO cells, we performed RAD (Regions Associated DEGs, https://labw.org/rad) analysis[34]_ENREF_33. We observed hyper-CG-DMR significantly positively corrected with downregulated genes in TCKO when the hyper-CG-DMR was located close to the promoters or potential enhancers of the genes (upstream 25 kb to downstream 25 kb of the transcription

start sites (TSS) (Supplementary Fig. 6a). GO term analysis of those downregulated genes with a hyper-CG-DMR within ± 2 kb of their TSS also showed strong enrichment over bone related GO terms such as ossification (Supplementary Fig. 6b). To further explore potential transcriptional factors binding over hyper-CG-DMR, we performed known motif search over hyper-CG-DMR. Top five enriched known motifs included bZIP, Runt, CTF, bHLH, and TEA/TEAD families (Supplementary Fig. 6c). Considering that the TET proteins can catalyze the oxidation of 5mC into 5hmC, we also examined the genome-wide 5hmC levels in WT and TCKO osteoblasts using 5hmC-Seal (Supplementary Fig. 5i). And DNA methylation in different organisms has been linked to chromatin structure[35]. We thus performed ATAC-seq and further analyzed the alteration of chromatin accessibility in TCKO cells. Consistent with the WGBS data, we observed genome-wide reduction of chromatin accessibility over protein-coding genes in TCKO cells compared to WT (Supplementary Fig. 6d). Plotting 5hmC-Seal and ATAC-seq signals over hyper-CG-DMRs, we observed a significant reduction in 5hmC level and chromatin accessibility in TCKO cells (Fig. 5a). Given that regulatory elements are typically hypomethylated and open compared to the surrounding chromatin structures[36], we hypothesized that the change of DNA methylation and chromatin accessibility level in TCKO may affect the binding of certain transcription factors. We thus identified chromatin regions that become more open or closed in TCKOs (Supplementary Fig. 6e). Overall, we detected around twofold more Open-to-Close-Region (oC-R, n = 39643) than Close-to-Open-Region (cO-R, n = 13302) in TCKO cells (Supplementary Fig. 6e), while most of TCKO oC-R enriched over promoter, 3'UTR, intron, and transcription termination sites (TTS) (Supplementary Fig. 6f). Motif analysis over TCKO oC-R identified CTF, Runt, Homeobox, bZIP, and TEA/TEAD families as the top five enriched known motifs (Supplementary Fig. 6g). TEA/TEAD transcriptional factors serve as the major partners of YAP1 and have been reported to inhibit osteogenesis before[37]. We also tested other potential transcriptional factors, including *Atf3*, *Nf1*, and *Pitx1*. We found that loss of *Atf3* and *Nf1* could inhibit osteogenesis determined by ALP activity and osteogenic markers (Supplementary Fig. 7a–d). Additionally, *Pitx1* showed little effects on osteoblast differentiation (Supplementary Fig. 7a–d).

To further reveal the relationship between the chromatin state change and TET-related DNA demethylation, we plotted the WGBS, 5hmC-Seal, and ATAC-seq signals over oC-Rs and cO-Rs. As shown in Supplementary Fig. 8, we observed stronger enrichment of differential CG methylation and 5hmC level in oC-R when compared to cO-Rs. These results support that TET-related CG demethylation promotes bone formation through the alteration of the chromatin state. We further analyzed the overlap between hyper-CG-DMR and chromatin oC-R in TCKO cells, and identified a significant amount of chromatin oC-R (10030/39643, 25.3%) overlapped with hyper-CG-DMR in TCKO cells (Fig. 5b). The results indicated that the dysregulation of gene expression may be closely related to the alteration of DNA methylation, and chromatin accessibility in TCKO cells. We next searched for motifs over regions shared by hyper-CG-DMR and chromatin oC-R in TCKO cells and identified Runt as the most enriched known motif with RUNX2 as one of the potential transcription factors (TFs) (Fig. 5c). RUNX2 is the key regulator of bone formation, and *Prx1-Cre, Runx2^{f/f}* mice displayed the shortness of the limbs and the dwarfism[38]. These results led us to investigate the potential relationship between RUNX2 and TET proteins. We identified RUNX2 binding sites from a published RUNX2 ChIP-seq dataset[39] and analyzed the CG methylation level and ATACs-seq signals over those sites in our TCKO cells. We observed obvious hyper CG methylation and hypo ATAC signals over RUNX2 ChIP-seq peak regions in TCKO cells (Fig. 5d). Based on these findings, we then identified differentially expressed genes associated with both RUNX2 and hyper-CG-DMR with RAD analysis (Fig. 5e). We observed strong enrichment of

downregulated genes with overlapping regions of RUNX2 binding sites and hyper-CG-DMRs, especially if those regions appeared in their promoters and potential enhancers (Fig. 5e). GO term analysis over those genes indicated ossification as the most enriched terms suggesting those genes would likely be the potential direct targets of RUNX2 and TET1/2/3 (Fig. 5f).

Then we focused on several RUNX2-regulated genes, including *Alpl*, *Col1α1*, and *Col1α2*. These genes were decreased in primary osteoblasts isolated from both TCKO and *Runx2*-heterozygous knockout mice, respectively (Supplementary Fig. 9a, b). A screenshot of genomics data indicated that RUNX2 was enriched over those marker genes including *Alpl*, *Col1α1*, and *Col1α2* (Supplementary Fig. 9c–e). We observed a consistent pattern showing the binding of RUNX2, enhancement of 5mC over promoters of potential enhancers in TCKO and less open chromatin in TCKO over those marker genes (Supplementary Fig. 9c–e). To further validate our results, we applied Bisulfate Sanger Sequencing and observed consistent hypermethylation over the promoter regions of *Alpl*, *Col1α1*, and *Col1α2* in TCKO cells (Supplementary Fig. 9f–h). Except for these marker genes, we also discovered a bunch of genes from the association analysis of hyper CG methylation with DEGs in TCKO cells, including *Ctsk*, *Fgf7*, *Clec11a*, and *Cys1* (Supplementary Table 1). *Ctsk* has been reported to be a periosteum stem cell marker[40], and loss of *Lkb1* in *Ctsk*-positive cells will result in osteosarcoma[41]. *Ctsk* can prevent lactation-induced bone loss in osteocytes[42]. *Fgf7* can facilitate osteoblast differentiation and bone formation through the activation of RUNX2 signaling[43,44]. *Clec11a* is also proved to be essential for osteogenesis[45]. *Cys1*, a cilia-associated protein, has not been investigated in skeletal system. These genes were hypermethylated and in a closed chromatin status in TCKO cells, compared with WT cells (Supplementary Fig. 10a–e). Interestingly, the promoter regions of these genes were also bound by RUNX2 (Supplementary Fig. 10a–e). These data further suggested the potential association of RUNX2 and TET enzymes in regulating specific gene expression.

## RUNX2 recruits TETs to its target genes to facilitate gene expression

From the motif analysis over hyper-CG-DMR in TCKO, we identified the top five enriched known motifs included Runt, CTF, bZIP, TEA/TEAD, RHD, and bHLH families. To investigate if there exists an interaction between TET enzymes and these transcription factors, we performed immunoprecipitation with TET2 and these candidates. We found that among those candidates' binding partners, RUNX2 could directly interact with TET2 (Supplementary Fig. 11a, b). Furthermore, all three TET proteins could directly interact with RUNX2 (Fig. 6a). The catalytic domain, but not the N-terminal domain of TETs can interact with RUNX2 (Fig. 6b and Supplementary Fig. 12a). GST pulldown experiments further confirmed the direct interaction between TET2 catalytic domain and RUNX2 (Supplementary Fig. 11b). Additionally, we found that TET1-CD and TET2-CD can interact with RUNX2 in C3H10 cells (Supplementary Fig. 11c). Functionally, coexpression of TET2 and RUNX2 could synergistically promote osteoblast differentiation, determined by increased ALP activity at day 7 (Fig. 6c left and d) and calcium nodule formation at day 21 (Fig. 6c right). This synergistic effect was further demonstrated by the increased expression of a series of osteogenic marker genes, including *Col1α1*, *Alpl*, and *Bglap* (Fig. 6e–i). Overall, our data demonstrated that the interaction between TET and RUNX2 synergistically promotes osteogenesis.

We next sought to investigate whether TET proteins can regulate the transcriptional activity of RUNX2. Osteocalcin-specific element 2 (OSE2) is a conserved short motif for RUNX2 in the promoter regions of a series of osteogenic genes[46,47]. We constructed a luciferase reporter in which 6xOSE2 element was linked with a DMR element from a *Col1a1* promoter (6xOSE2 + DMR) (Fig. 7a). In the absence of

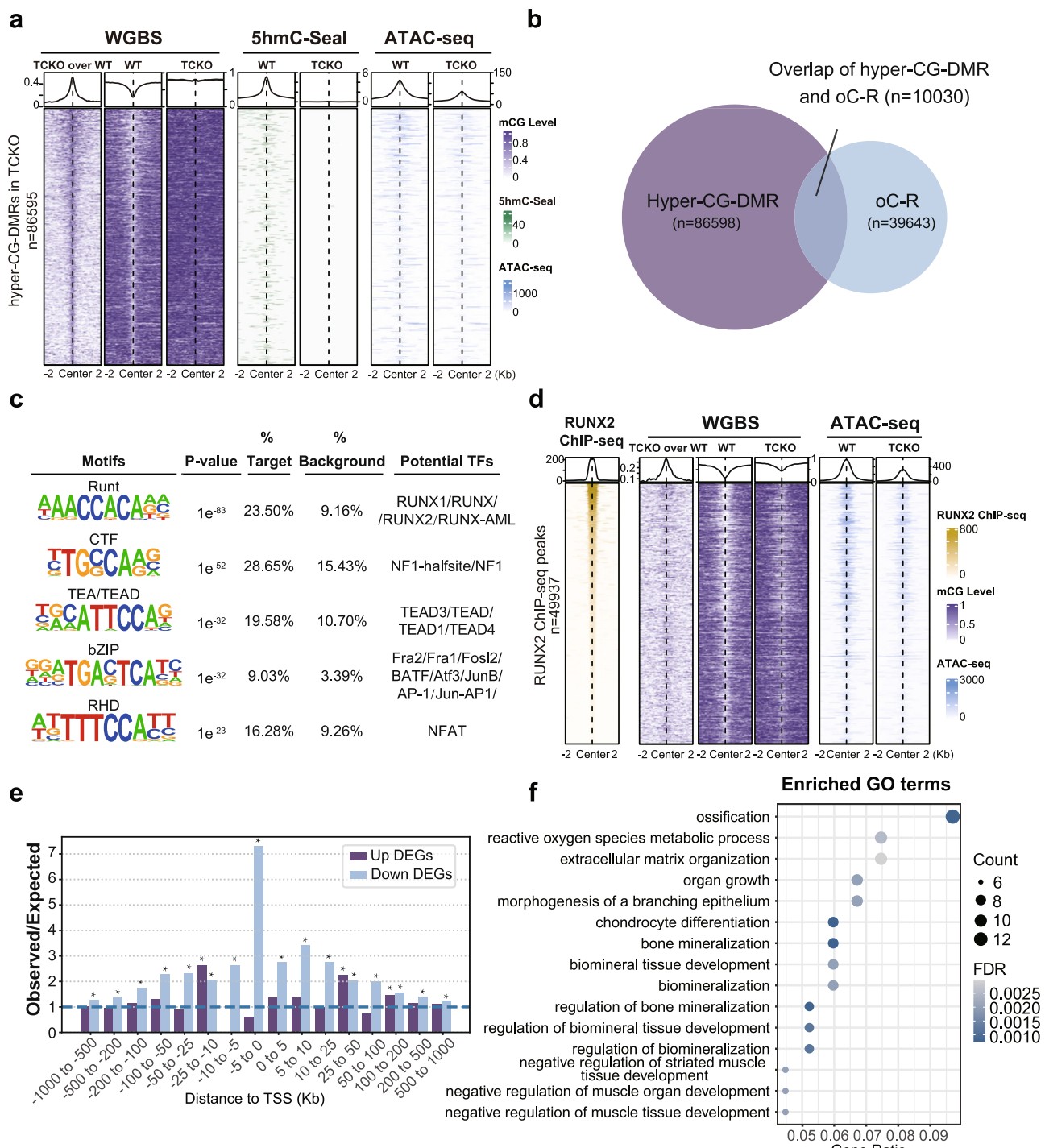

**Fig. 5 | Association analysis of RNA-seq, WGBS, 5hmC-Seal, ATAC-seq, and RUNX2 ChIP-seq data suggests TET and RUNX2 proteins could synergistically promote osteogenesis. a** Heatmap and metaplot of CG methylation level, 5hmC-Seal, and ATAC-seq signals over hyper-CG-DMR in TCKO cells. **b** Venn diagram of hyper-CG-DMR and chromatin Open-to-Close-Region (oC-R) in TCKO cells. **c** Top five enriched known motifs and their potential transcription factors in overlapped regions of hyper-CG-DMR and chromatin oC-R in TCKO cells. **d** Heatmap and metaplot of RUNX2 ChIP-seq, CG methylation level, and ATAC-seq signals of TCKO and WT cells over RUNX2 binding sites. **e** RAD analysis for regions shared by hyper-CG-DMRs in TCKO cells and RUNX2 binding sites and DEGs in TCKO cells compared with WT. *$P < 0.01$, the exact $P$ values were provided in the source data; hypergeometric test. **f** Enriched GO terms of downregulated genes in TCKO cells, which had a distance less than 100 kb from overlapped regions of hyper-CG-DMRs in TCKO cells and RUNX2 binding sites.

TET proteins, DMR elements can inhibit the RUNX2 activation on 6xOSE2 (Fig. 7b, c). Strikingly, the inhibition by DMR elements can be released by TET2 (Fig. 7d). Using ChIP-qPCR, we observed the reduction of RUNX2 enrichment in the promoter regions of osteogenic marker genes including *Col1α1*, *Col1α2*, and *Alpl* in TCKO osteoblasts, compared with the wild type osteoblasts (Fig. 7e–g). Consistently, RNA

polymerase II (RNA Pol II) binding over those regions also decreased in the TCKO osteoblasts (Fig. 7h–j). Combined with the increased methylation and closed chromatin accessibility in TCKO osteoblasts (Supplementary Fig. 9c–h), we summarized the model that TETs and RUNX2 can form a complex to mediate the demethylation of RUNX2 target genes. Loss of TETs can give rise to the hypermethylation and

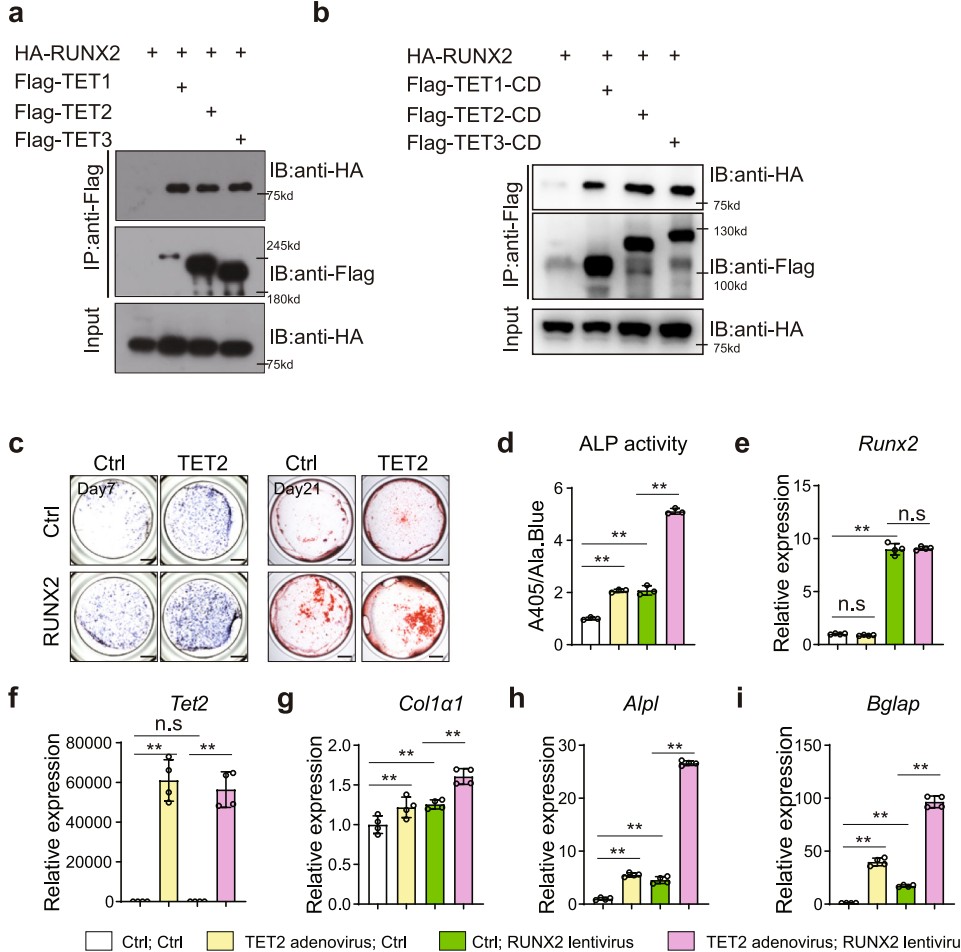

**Fig. 6 | TET proteins associate with RUNX2 and promote osteogenesis with RUNX2 synergistically. a** Co-immunoprecipitation (Co-IP) of Flag-TET1 or Flag-TET2 or Flag-TET3 with HA-RUNX2. HA-RUNX2 expressing plasmid was co-transfected with Flag-TET1 or Flag-TET2 or Flag-TET3 in 293 T cells. Whole cell lysate was used for immunoprecipitation and then blotted with indicated antibodies. Representative images for two independent samples. **b** Co-immunoprecipitation (Co-IP) of Flag-TET1-CD or Flag-TET2-CD or Flag-TET3-CD with HA-RUNX2. HA-RUNX2 expressing plasmid was co-transfected with Flag-TET1-CD or Flag-TET2-CD or Flag-TET3-CD in 293 T cells. Whole cell lysate was used for immune precipitation and then immunoblotting with indicated antibodies.

Representative images for 3 independent samples. **c** ALP staining and Alizarin red S staining after osteoblast differentiation for 7 days (left) and 21 days (right), respectively. Cells were infected with *Runx2*-lentivirus and *Tet2*-adenovirus separated or combined. Scale bar = 1 mm. **d** ALP activity quantification was measured by phosphatase substrate assay as A405/Ala.Blue. *$P < 0.05$, **$P < 0.01$. Ordinary one-way ANOVA. Data are presented as mean ± s.d., $n = 3$ independent cell supernatants. **e**–**i** RT-qPCR analysis of *Runx2* (**e**), *Tet2* (**f**), *Col1α1* (**g**), *Alpl* (**h**), and *Bglap* (**i**) expression after osteoblast differentiation for 7 days, cells were from WT and TCKO mice. *$P < 0.05$, **$P < 0.01$. Ordinary one-way ANOVA. Data are presented as mean ± s.d., $n = 4$ independent samples.

closed accessibility of chromatin, and then inhibit the binding and transcription of RUNX2 (Fig. 7k).

## Discussion

Our current study demonstrates that loss of all three TET proteins in mesoderm mesenchymal stem cells results in severe bone defects and any single allele of *Tet* genes can rescue the defects of bone development. Another investigation demonstrated that *Tet1*, *Tet2*, and *Tet3* were all expressed in both human and mouse BMMSCs, but only *Tet1* and *Tet2* decreased in the ovariectomized (OVX) mice-derived BMMSCs[21]. In our study, *Tet1* and *Tet2* double conditional knockout mice showed subtle differences, compared with WT mice at P0, indicating that during the embryonic bone formation *Tet3* may provide compensated function in the absence of *Tet1* and *Tet2*. These observations indicate the functional compensation among three TET proteins during the skeletal developmental process, similarly to the process of early gastrulation[10].

TET proteins have dual functions in the regulation of gene expression. TET-mediated demethylation in the promoter region is correlated with active gene expression. Additionally, TET proteins

involve gene silencing by binding to certain regulatory factors and protein complexes. For example, TET proteins can interact with chromatin repressive complexes like PRC2 and SIN3A to repress gene repression[48,49]. The up-regulated genes in the TCKO cells are likely regulated in this manner. The neuroactive ligand-receptor interaction and axon guidance pathway were enriched in up-regulated genes of TCKO cells. Among the up-regulated genes, osteoblast-derived *Cox2* has been reported to promote the innervation of sensory nerve fibers[50]. The sensory nerves can sustain the bone mass accrual[51]. This could be a compensated result of bone loss in TCKO mice.

The transcriptional activity and posttranslational regulation of RUNX2 are widely investigated since it is the master of osteogenesis. Mutation and dysregulation of RUNX2 is a key factor giving rise to human CCD, an autosomal-dominant skeletal disease that is typically characterized by delayed closure of calvarial fontanels and clavicle hypoplasia[17,18,52–54]. Previously studies have reported that several factors can modulate RUNX2 activity. TAK1−MKK3/6−p38 MAPK axis promotes association between RUNX2 and the co-activator CREB-binding protein (CBP) by phosphorylating RUNX2, which is required in osteoblast genetic programs[55]. Transcriptional complex mediator

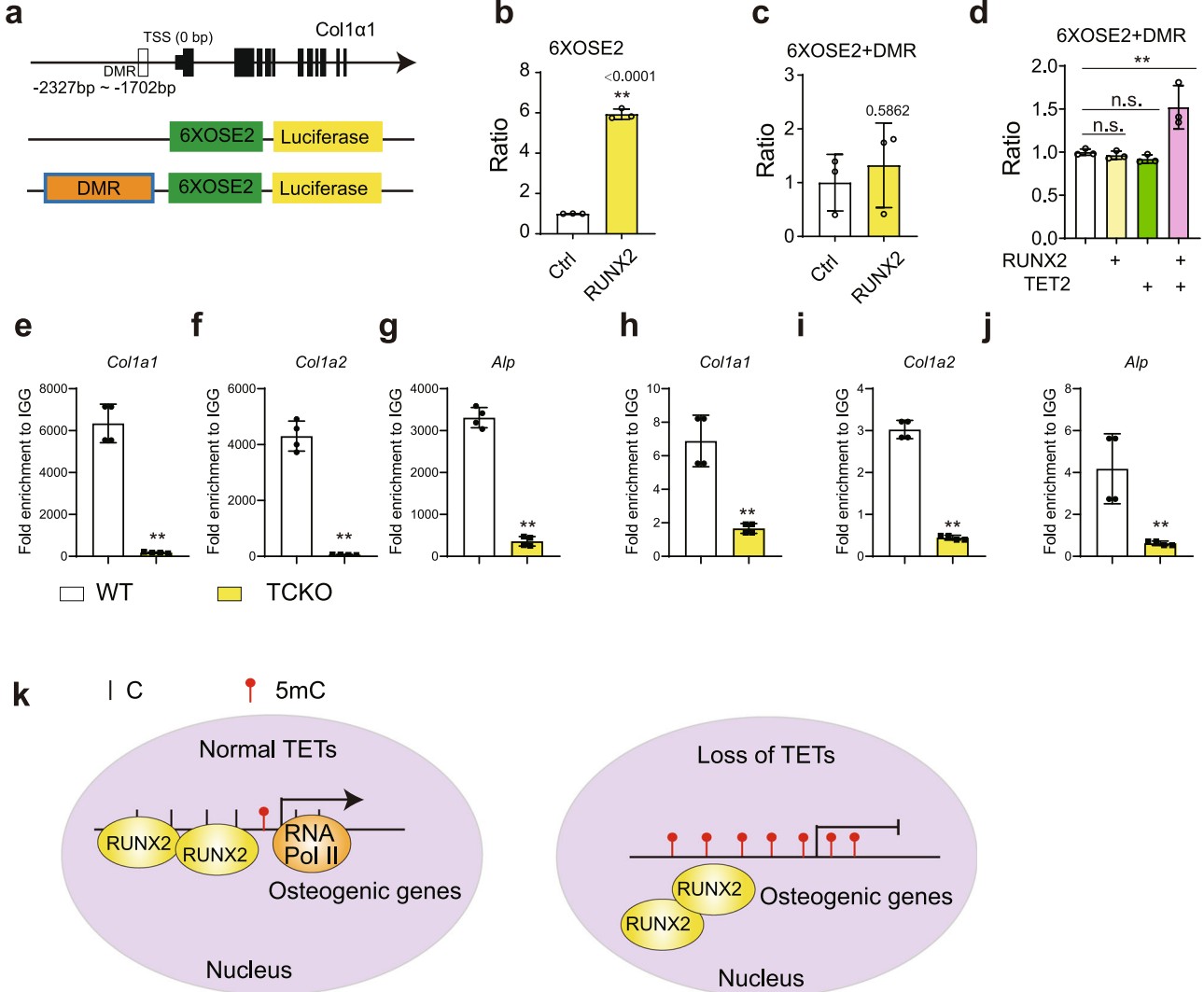

**Fig. 7 | RUNX2 recruits TET proteins to the promoter of target genes to facilitate transcription by modification of DNA methylation. a** The strategy for promoter luciferase assay. OSE2: Osteocalcin-specific element 2. DMR: −2327bp to −1702bp of *Col1α1* promoter. **b**, **c** The measurement of luciferase activity for 6XOSE2 (**b**) and 6XOSE2 + DMR (**c**) promoter stimulated by RUNX2. **\*\****P* < 0.01. Two-tailed Student's *t* test. Data are presented as mean ± s.d., *n* = 3 independent cell samples. **d** The measurement of luciferase activity for 6XOSE2 + DMR treated with TET2 or RUNX2 separated or combined. **\*\****P* < 0.01. Ordinary one-way ANOVA. Data are presented as mean ± s.d., *n* = 3 independent cell samples. **e**–**g** ChIP-qPCR for RUNX2 over the promoter regions of osteogenic genes including *Col1α1* (**e**), *Col1α2*

(**f**), *Alpl* (**g**). **h**–**j** ChIP-qPCR for RNA Polymerase II (RNA Pol II) over the promoter regions of osteogenic genes including *Col1α1* (**h**), *Col1α2* (**i**), *Alpl* (**j**). **\*\****P* < 0.01. Two-tailed Student's *t* test. Data are presented as mean ± s.d., *n* = 4 independent samples. **k** Based on our results, we proposed the model that RUNX2 could bind to the promoter regions of osteogenic genes. By recruiting TET proteins via RUNX2, DNA hypomethylation would be induced over those osteogenic genes and thus promote their expression by providing a more accessible chromatin landscape for transcriptional machinery such as RNA Pol II. Loss of all TET proteins would result in repression of osteogenic genes targeted by RUNX2, and thus lead to bone development failure.

Med23 binds to RUNX2 and enhances RUNX2 transcriptional activity[54]. MiRNA-133a-5p have been proposed to inhibit RUNX2 expression by targeting its 3′-UTR[56]. Recently, Shi group demonstrated that loss of *Tet1* and *Tet2* results in osteoporosis[21]. In their study, the expression of *Runx2* was reduced by the accumulation of miR-297a-5p, miR-297b-5p, and miR-297c-5p in *Tet1/2* double knockout BMSCs. However, the methylation status of these miRNAs was not affected in *Tet1/2* double knockout BMSCs. Interestingly, P2X purinoceptor 7 (*P2rX7*) expression, capable of promoting the release of exosomes containing these miRNAs, was repressed by DNA hypermethylation in the promoter region due to loss of both TET1 and TET2[21,57]. Consistently, our WBGS results showed that *P2rX7* promoter was hypermethylated in *Tet*-deficient osteoblasts, further confirming that *P2rX7* can be regulated by TET-mediated DNA hypomethylation (data not shown).

In this study, we mainly focus on the regulation of RUNX2 target genes in osteoblasts by TET proteins. Interestingly, COl10a1 and MMP13 in chondrocytes were also affected in TCKO mice (Supplementary Fig. 3b–d), indicating that DNA methylation also played key roles in the regulation of the genes critical for endochondral ossification. We analyzed the COl10a1 and MMP13 in the high-throughput sequencing data of the primary osteoblast cells for some clues. Both COl10a1 and MMP13 were occupied by RUNX2 (Supplementary Fig. 10f, g). Loss of TET proteins decreased the chromatin accessibility of COl10a1 and MMP13 in osteoblasts with the increased 5mC and decreased 5hmC level (Supplementary Fig. 10f, g). Although we did not use primary chondrocytes for the study, these data indicated that COl10a1 and MMP13 might also be regulated in chondrocytes by the interaction of TET proteins and RUNX2. However, the detailed mechanism of how TET proteins and RUNX2 collectively regulated the

gene expression of COl10a1 and MMP13 is worthy to be further investigated.

The clavicle is formed by both intramembranous and endochondral ossification, which is a unique bone formation pathway. The defects of the clavicle are mostly caused by the RUNX2 and its coregulators, including Med23[47], Vgll4[37], etc, which play essential roles in the regulation of both intramembranous and endochondral ossification. As TET proteins could interact with RUNX2 and regulate the RUNX2 targeted gene expression, and RUNX2 could affect both endochondral and intramembranous bone formation, we supposed that the shorter clavicles in TCKO mice might be caused by the dysregulated RUNX2 targeted gene expression by loss of TETs proteins.

In conclusion, our study found that TET family-mediated DNA demethylation was indispensable for maintaining embryonic bone formation. *Tet1*, *Tet2*, and *Tet3* are functionally redundant for this process. The specific DNA demethylation of TET proteins on osteogenic genes required the RUNX2 guidance, and in turn facilitated the expression of these targeted genes. Our investigation expands the understanding of DNA methylation on RUNX2-regulates bone development and advances the clinical diagnosis.

## Methods

### Mouse lines

*Tet1*<sup>fl/fl</sup> *Tet2*<sup>fl/fl</sup> *Tet3*<sup>fl/fl</sup> mice and *Tet1*<sup>fl/HD</sup> *Tet2*<sup>fl/fl</sup> *Tet3*<sup>fl/fl</sup> mice (gifts from Guoliang Xu, Shanghai Institute of Biochemistry and Cell Biology)[10] were crossed with the *Prx1-Cre* strain[22] (a gift from Andrew McMahon, Harvard University) to generate *Prx1-Cre, Tet1*<sup>fl/fl</sup> *Tet2*<sup>fl/fl</sup> *Tet3*<sup>fl/fl</sup> mice, *Prx1-Cre, Tet1*<sup>fl/+</sup> *Tet2*<sup>fl/fl</sup> *Tet3*<sup>fl/fl</sup> mice, *Prx1-Cre, Tet1*<sup>fl/fl</sup> *Tet2*<sup>fl/+</sup> *Tet3*<sup>fl/fl</sup> mice, *Prx1-Cre, Tet1*<sup>fl/fl</sup> *Tet2*<sup>fl/fl</sup> *Tet3*<sup>fl/+</sup> mice, *Prx1-Cre, Tet1*<sup>fl/HD</sup> *Tet2*<sup>fl/fl</sup> *Tet3*<sup>fl/fl</sup> mice and *Prx1-Cre, Tet1*<sup>+/HD</sup> *Tet2*<sup>fl/fl</sup> *Tet3*<sup>fl/fl</sup> mice. All mice analyzed were maintained on the C57BL/6 background. Animals were euthanized by $CO_2$. The experiments were mainly performed with newborn mice or embryonic mice. The specific detailed age was noted in the figure legend. Animals were bred and maintained under specific pathogen-free conditions with the 12/12 hours light/dark cycle, 25°C, and 45-65% humidity in the institutional animal facility of the Shanghai Institute of Biochemistry and Cell Biology, Chinese Academy of Sciences. All experiments were performed with a protocol approved by the Animal Care and Use Committee of Shanghai Institute of Biochemistry and Cell Biology, Chinese Academy of Sciences.

### Cell culture

Primary osteoblasts were isolated from the calvarial bone of postnatal day 3–5 mice. Parietal bone was separated from skull and cut into pieces. These pieces of bone were digested by α-MEM with 1 mg/ml collagenase (Sigma, C0130) and 2 mg/ml Dispase II (Sigma, D4693) for 10 minutes per time four times. The last three digested cells were collected into 15 ml centrifuge tube containing α-MEM (Corning) supplemented with 10% fetal bovine serum (FBS) and centrifuged at 1200 g for 3 min. The cell pellet was re-suspended with α-MEM with 10% FBS and 1% penicillin/streptomycin and plated into 10 cm dishes.

For osteoblast differentiation, cells were induced by α-MEM with 10% FBS, 1% penicillin/streptomycin, 50 μg/ml L-ascorbic acid (Sigma, A5960), and 1.08 mg/ml β-Glycerophosphate disodium salt hydrate (Sigma, G9422). The osteoblast differentiation level assay was performed following a previously published method[58]. For ALP staining, cells were fixed with 4% paraformaldehyde for 10 min on day 7 and then stained according to the manual of BCIP/NBT Alkaline Phosphatase Color Development Kit (Beyotime, C3206). For quantitative analysis of ALP activity, osteoblasts were incubated with Alamar Blue (Thermo, 88951) for 1 hour for cell number quantification. Then the cells were incubated in a solution (6.5 mM $Na_2CO_3$, 18.5 mM $NaHCO_3$, 2 mM $MgCl_2$, and phosphatase substrate (Sigma, S0942) for 20 min. Alamar Blue and ALP activity were then read by a luminometer

(Envision). Bone nodule formation was stained with 1 mg/ml alizarin red s solution (pH 5.5) after 21 days of induction.

Primary chondrocytes were isolated from articular cartilages of postnatal day 3–5 mice. The articular cartilages were digested by α-MEM with 1 mg/ml collagenase and 2 mg/ml Dispase II for 10 min to remove the fibroblasts and muscle cells, then for 1 hour to collect the cells. For chondrocyte differentiation, a drop (12.5 μl, $2.5 \times 10^5$ cells) chondrocytes were seeded in the 24-well plate. After 2 hours, 500 μl chondrogenic medium was added to the plate for 3 days and then the cells were stained with Alcian Blue or lysed for RNA extraction.

### Immunoprecipitation

Human embryonic kidney cell line (HEK293T) was obtained from ATCC and maintained in DMEM (Corning) containing 10% FBS and 1% penicillin/streptomycin. HEK293T were transfected with indicated plasmid (Flag-TET1: full length; Flag-TET2: full length; Flag-TET3: full length; Flag-TET1-CD:1367aa-2008aa; Flag-TET2-CD, 1042aa-1913aa; Flag-TET3-CD:697aa-1669aa; Flag-TET1-Nterm,1aa-1366aa; Flag-TET2-Nterm, 1aa-1041aa; Flag-TET3-Nterm:1aa-696aa). After 48 hours, cells were lysed with 1 ml EBC buffer (50 mM Tris, pH 7.5, 120 mM NaCl, and 0.5% NP-40) supplemented with a protease inhibitor cocktail (MCE, HY-K0010) and centrifuged at 12,000 rpm. 50 μl supernatant was kept as input samples, and 900 μl supernatant was incubated with 7 μl anti-Flag beads (Sigma, A2220) or anti-HA beads (AOGMA, AGM90054) for 3–4 hours or overnight at 4 °C. Anti-Flag beads or HA beads serve as the control to rule out non-specific binding. In the absence of Flag-tagged protein, anti-Flag beads could not pull HA-tagged proteins down. In the same way, anti-HA beads could not pull Flag-tagged proteins down in the absence of HA-tagged protein. Immunoprecipitated targets were washed five times with the EBC buffer before analysis by sodium dodecyl sulfate–polyacrylamide gel electrophoresis (SDS-PAGE) and immunoblotted with primary antibodies including HA antibody (Santa Cruz Biotechnology, sc-7392, 1:2000), and Flag antibody (Sigma, F3165-5MG, 1:5000) and secondary antibody of Anti-mouse-immunoglobulins/HRP (Abbkine, A2502, 1:2000).

### Histology and immunochemistry

Tissues were fixed in 4% paraformaldehyde for 48 h and incubated in 15% DEPC-EDTA (PH 7.8) for decalcification. Then specimens were embedded in paraffin and sectioned at 6 μm. In brief, the samples were immersed in xylene, ethanol, 70% ethanol, and water for dewaxing. For Safranin O staining, the dewaxing samples were immersed in 0.05% Fast green for 3 min, 1% acetic acid for 10 s, 0.5% Safranin O for 5 min, 95% ethanol for 10 s, and water for 5 min. For Von-kossa staining, the dewaxing samples were immersed in silver nitrate for 5 min, water for 5 min for three times, 5% sodium thiosulfate for 5 min, water for 5 min for three times, and Nuclear fast red for 5 min, water for 5 min for three times. Slides were mounted with neutral balsam. Images were taken using a microscope (Olympus BX51, Tokyo, Japan). For TRAP staining, the slides were immersed in the staining buffer at 37°C for 20 min. The staining buffer was a mixture of A buffer (125 ul 10 mg/ml Naphthol AS-MX in 12.5 mL 0.1 M acetate buffer), B buffer (125 μl 15 mg/ml Fast Red LB in 12.5 mL 0.1 M acetate buffer) and 1.5 mL 1 M Sodium Tartrate buffer. For the Edu staining, 0.05 mg/g Edu was injected into the mice at P0. After 24 hours, the mice were sacrificed and the femurs were fixed, decalcified, and embedded in paraffin. The slides were stained using the Cell-Light Apollo 488 stain kit (RIBOBIO, C10371-3).

### Immunofluorescence

Freshly dissected bones were fixed in 4% paraformaldehyde for 48 h and incubated in 15% DEPC-EDTA (PH 7.8) for decalcification. Then specimens were embedded in paraffin or sectioned at 6 μm. Sections were blocked in PBS with 10% horse serum and 0.3% Triton X-100 (Sangon Biotech, A110694-0500) for 1 hour and then stained overnight with goat-anti-OPN (R&D Systems, AF808, BD00720111, 1:1000;),

rabbit-anti-BGLAP (ABCAM, ab93876, GR3357172-2,1:200), rabbit-anti-RUNX2 (ABCAM, ab236639, GR3388032-6,1:200), mouse-anti-5mC (ABCAM, ab10805, GR284756-4, 1:1000), rabbit-anti-5hmC (active motif, 39792,1:1000), rabbit-anti-Col10a1 (Abclonal, a6889, 3561636001, 1:200), rabbit-anti-Mmp13 (Proteintech, (Proteintech, 18165-1-AP, 021, 1:200), rabbit-anti-Osx (ABCAM, ab22552, GR3357012-2, 1:200), biotin-anti-PCNA (Biolegend, 307904, B213618, 1:200), or rabbit-anti-Cleaved-Caspase3 (CST, 9664 T, 21, 1:200). Donkey-anti-mouse 488 (Molecular Probes, A21202, 1:1000), donkey-anti-goat 488 (Molecular Probes, A11055, 1:1000), donkey-anti-rabbit cy3 (Jackson ImmunoResearch, 711-165-152, 1:1000) or APC-Streptavidin (BioLegend, 405207, B266711, 1:500) were used as secondary antibodies. DAPI (Sigma, D8417) was used for counterstaining. Slides were mounted with an anti-fluorescence mounting medium (Dako, S3023) and images were captured with a Leica SP5 and SP8 confocal microscope.

### Real-time RT-PCR analysis

Total RNA was prepared using TRIzol (T9424, Sigma) and was reverse transcribed into cDNA with the PrimeScript RT Reagent Kit (PR037A, TakaRa). The real-time reverse transcriptase (RT)-PCR reaction was performed with the BioRad CFX96 system. The primer sets were used as follows:

mTET*1*-qF: CATTCTCACAAGGACATTCACAACA;
mTET*1*-qR: AGTAAAACGTAGTCGCCTCTTCCTG;
mTET*2*-qF: TATTGAGGGTGACCACCACTGTACT;
mTET*2*-qR: GCTCCAATATACAAGAAGCTTGCAC;
mTET*3*-qF: CTATCAGAACCAGGTGACCAATGAG;
mTET*3*-qR: ACAGTGCACCCATTGTAGAGGTTAT;
mRUNX2-qF: TTTAGGGCGCATTCCTCATC;
mRUNX2-qR: TGTCCTTGTGGATTAAAAGGACTTG;
m*Sp7*-qF: CTCTGCTTGAGGAAGAAGCTCAC;
m*Sp7*-qR: CTTCTTTGTGCCTCCTTTCCC;
m*Alpl*-qF: CGGGACTGGTACTCGGATAA;
m*Alpl*-qR: ATTCCACGTCGGTTCTGTTC;
m*Col1α1*-qF: GCTCCTCTTAGGGGCCACT;
m*Col1α1*-qR: CCACGTCTCACCATTGGGG;
m*Col1α2*-qF: CCTGGTCCTCACGGTTCTG;
m*Col1α2*-qR: ACCCTGCAATCCACTGTATCCT;
m*Bglap*-qF: CTTGGTGCACACCTAGCAGA;
m*Bglap*-qR: CTCCCTCATGTGTTGTCCCT.

### ChIP assays

Chromatin immunoprecipitation (ChIP) assays were performed with the Magna ChIP A/G kit (Millipore, 17-10085) following the manufacturer's instructions. A total of $1 \times 10^6$ osteoblast cells isolated from WT and TCKO mice were fixed in 1% formaldehyde for 10 minutes and quenched unreacted formaldehyde by glycine for 5 minutes followed by two times washing. After scraped and collected, cells were re-suspended in cell lysis buffer containing 1× protease inhibitor cocktail II and sonicated. Centrifuged to obtain cell extracts, which then were incubated overnight at 4 °C with protein A/G magnetic beads and RUNX2 (ABCAM, ab236639, 3 μg/$1 \times 10^6$ cells) or Pol II (Millipore, 05-623, 3 μg/$1 \times 10^6$ cells) antibodies respectively. Normal IgG (Millipore, 12-371, 3 μg/$1 \times 10^6$ cells) was used as a negative control. Subsequently, the protein/DNA complexes were eluted and reversed cross-links. Finally, the purified DNA was used to qPCR to amplify various regions of the target gene genome. Normal IgG was used as a negative control. qPCR was used to amplify various regions of the target gene genome. Primers for ChIP-qPCR:

*Col1a1*-F: TCCTTTCAAACCACCACATA;
*Col1a1*-R: GCATCCTGGGGAAGAACTAA;
*Col1a2*-F: AGTCTGGCTGGTAATCCTTCAT;
*Col1a2*-R: CCTCCCTCCTCCATCCTCT;
*Alpl*-F: CCGAAGTCACCAGAGCAGG;
*Alpl*-R: AAGAAACAGGGACTCAGGACA.

### Skeletal preparation and staining

Mice were sacrificed by $CO_2$, and then the skin and gut were removed. The remaining part was transferred into acetone for 48 h after overnight fixation in 95% ethanol. Skeletons were then stained in Alcian blue and Alizarin red S solution (0.3% Alcian blue in 70% ethanol, 10 ml; 0.1% Alizarin red S in 95% ethanol, 10 ml; acetic acid, 10 ml; 170 ml 70% ethanol) for 2 or 3 days[59]. Specimens were kept in 1% KOH until tissue became clear.

### Conventional bisulfite sequencing

The genomic DNA was treated with the EZ DNA Methylation-Direct Kit (Zymo Research, D5005) according to the manufacturer's instructions. Bisulfite-treated DNA was subjected to PCR amplification. The bisulfite primers are designed by an online tool, The Li Lab (http://www.urogene.org/cgi-bin/methprimer/methprimer.cgi)[60]. PCR products were purified with a Gel Extraction Kit (Hling, HDP202-02), and cloned into pGEM-T Easy Vector (Promega, A1360). Individual clones were sequenced by standard Sanger Sequencing. Data were analyzed by an online tool BISMA (http://services.ibc.uni-stuttgart.de/BDPC/BISMA/)[61].

### In vitro binding assays

Briefly, GST-RUNX2 proteins were expressed in the BL21(DE3)pLysS *E. coli* strain followed by purification using Glutathione Sepharose 4B (GE Healthcare, 17-0756-01) according to the manufacturer's instruction. The Flag-TET2-CD protein was purified by flag beads and eluted by flag peptide (Sigma). Purified GST and GST-RUNX2 proteins were incubated with Flag-TET2-CD protein in the EBC buffer for 3-4 h at 4 °C followed by washing with the EBC buffer three times before analysis by SDS-PAGE.

### Luciferase reporter assays

6XOSE2 reporter plasmid has been reported in our previously study[62]. DMR + 6XOSE2 reporter plasmid was constructed by inserting a DMR sequence (from −2327 bp to −1702 bp) of *Col1α1* promoter before 6XOSE2 sequence. For the luciferase reporter assay, HEK293T cells were transfected with indicated luciferase reporter plasmids and different combinations of expression constructs. All cells were co-transfected with Renilla (Promega) as a normalization control. 48 h after transfection, cells were harvested and lysed by EBC buffer. Luciferase assay was performed using the Dual-Luciferase Reporter Assay System (Promega, E1960).

### RNA sequencing, GBS, and 5hmC-Seal sequencing

Primary osteoblasts were isolated from 3-day-old WT and TCKO mice. After 3-day culture, cells were harvested and divided into two parts for RNA and DNA extraction. RNA was extracted by TRIzol (Sigma, T9424), while DNA was extracted by DNeasy Blood & Tissue Kit (QIAGEN, 69504). The RNA-seq library constructions and sequencing were utilized 3 repeats per group in WT and TCKO osteoblasts. Libraries were sequenced on Illumina X10 (The CAS-MPG Partner Institute for Computational Biology, Shanghai, China). For the WGBS, BS-seq library constructions and sequencing utilized one sample per group. Libraries were sequenced on the Illumina Hiseq 2500 platform (Novogene, Beijing, China). For the 5hmC-Seal sequencing, the library construction and sequencing were accomplished by the EPICAN (http://www.epican.cn/).

### Assay for Transposase-Accessible Chromatin (ATAC-seq)

A total of $1 \times 10^5$ osteoblasts from WT and TCKO mice were collected, washed in additional buffer (0.04% BSA in PBS), and lysed in 50 μl lysis buffer (10 mM Tris-HCl pH 7.4, 10 mM NaCl, 3 mM $MgCl_2$, 0.1% Tween-20, 0.1% Nonidet P40 Substitute, 0.01% digitonin and 1% BSA) for 5 min on ice. Then, 50 μl wash buffer (10 mM Tris-HCl, 10 mM NaCl, 3 mM MgCl2, 1% BSA, 0.1% Tween-20) was added to each tube followed by

centrifugation. The subsequent processes were carried out by True-Prep DNA Library Prep Kit V2 for Illumina (Vazyme, TD501) according to the manufacturer's instructions. In brief, the cell nuclei were fragmented and tagged by TN5 transposase for 30 min at 37 °C, and the DNA fragments were amplified by 6–9 PCR cycles. Subsequently, the library fragments were purified by DNA clean&Concentrator kit (Zymo, D4014) and VAHTS DNA clean beads (Vazyme, N411-00-AA), and finally sequenced.

### Micro-quantitative computed tomography analysis
Skulls were isolated from 10-day mice and fixed in 70%. Then the skull was scanned by the micro-CT (Scanco μct80) with a spatial resolution of 10 μm. The scans were reconstructed into two-dimensional coronal mode.

### Next generation sequencing data analysis
**RNA-seq analysis.** The RNA sequencing reads were aligned to GRCm38.99 *Mus musculus* reference genome with STAR(v2.7.0e)[63] using a supplied set of known transcripts in GTF format (RefSeq GRCm38.99; *Mus musculus*, Ensembl). Differentially expressed genes were calculated using DESeq2[64] with cutoff (Fold change greater than 2, FDR less than 0.05, and RPKM greater than 0.24 - the first quartile), and RPKM values were calculated with a custom R script. Once differential expressed genes were determined, R package clusterProfiler[65] was used for further analysis of GO terms and KEGG pathway.

**WGBS analysis.** For WGBS data analysis, raw reads were aligned to the reference GRCm38.99 *Mus musculus* genome using BSMAP (v 2.74)[66] by allowing up to 2 mismatches (-v 2), 1 best hit (-w 1) and aligning to both strands(-n 1). Methylation levels at each cytosine were then extracted with BSMAP (methratio_32unit_alt.py) scripts by allowing only unique mapped reads (-u). Methylation levels at each cytosine were calculated as #C/(#C + #T). DMRs were then defined with R package DMRcaller[67] over the whole genome at least four CG sites in 200 bins and minimal CG methylation level difference greater than 40% and P-value less than 0.05 and min Gap as 100 bp. In order to identify the genomic distribution of hyper-CG-DMR and predominant motifs in hyper-CG-DMR, HOMER(v4.10.4)[68] was applied to 200 bp around the middle points of hyper-CG-DMRs. The hyper-CG-DMRs associated with DEGs analysis (RAD) were performed with the RAD web application via https://labw.org/rad[34] to calculate the distance between hyper -CG-DMR and DEGs.

**ATAC-seq analysis.** ATAC-seq data were aligned to the reference GRCm38.99 *Mus musculus* genome using STAR(v2.7.0e)[63] by using EndToEnd alignEndsType and allowing only uniquely mapping reads with fewer than three mismatches and only 1 for maximum intron length. Peaks were called using MACS2 with call-summits method and other default parameters (v 2.1.6)[69]. To find peaks specific to one condition (e.g. TCKO), specific peaks to each state were defined with a two-fold relative enrichment cutoff after merging the replicates. Genomic distribution and motif analysis of specific peaks in TCKO was calculated using HOMER (v4.10.4)[68] with 200 bp around the middle points of peaks. To determine the overlapped regions of oC-R with hyper-CG-DMRs, we used the bedtools intersect tool[70].

**ChIP-seq and 5hmC-Seal analysis.** For ChIP-seq and 5hmC-Seal data analysis, raw reads were aligned to the reference GRCm38.99 *Mus musculus* genome using STAR(v2.7.0e)[63] with the same parameters of ATAC-seq analysis. Peaks were called using MACS2 with the call-summits method and other default parameters (v 2.1.6)[69]. ChIP-seq and 5hmC-Seal metaplots and heatmaps were generated using R package EnrichedHeatmap[71].

### Statistical analysis
Statistical analysis was performed using GraphPad Prism 8 software (GraphPad Software). Cell-based experiments were performed at least twice. Animals were randomized into different groups and at least three mice were used for each group unless otherwise stated. The data are presented as mean ± s.d. Two-tailed Student's *t* test was used to compare the effects of the two groups. One-way ANOVA was used to compare more than two groups. Statistically significant differences are indicated as follows: * for $p < 0.05$ and ** for $p < 0.01$.

### Reporting summary
Further information on research design is available in the Nature Research Reporting Summary linked to this article.

## Data availability
The data that support this study are available from the corresponding authors upon reasonable request. The RNA-Seq, WGBS, ATAC-Seq, and 5hmC-Seal data are deposited in the Gene Expression Omnibus (GEO) under accession GSE174048. The RUNX2 ChIP-Seq data is from publicly available datasets in GEO under accession GSM1027478. Source data are provided with this paper.

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

## Acknowledgements

This work was supported by the National Natural Science Foundation of China (NSFC) (81991512, 82102554) and the Strategic Priority Research Program of the Chinese Academy of Sciences (XDA16021300). W.Z. is a scholar of the National Science Fund for Distinguished Young Scholars (NSFC) (no. 81725010). This work was also supported by Shanghai Sailing Program (21YF1434500).

## Author contributions

L.W. and W.Z. conceived the study and wrote the manuscript. W.L. and D.R. analyzed all of the sequencing data and wrote the manuscript. L.W. and X.Y. performed the experiment cooperatively. R.S., H.D., W.S., and G.X. contributed ideas and reviewed the manuscript.

## Competing interests

The authors declare no competing interest.
