## [Peer Review File · Nature Communications]

REVIEWER COMMENTS

Reviewer #1 (Remarks to the Author):

In this study Zou and colleagues delete Tet enzymes in the mesenchymal stem cells using Prx1-Cre and find that it leads to defects in embryonic and adult bone development. They further show that deficiency of Tets leads to deregulation of osteogenic genes and increased DNA methylation, specifically at Runx2 binding sites, in osteoblasts. They show that Tets interact with Runx2 in 293T cells when overexpressed, and conclude that Tet enzymes facilitate Runx2 recruitment to its target osteogenic genes for their proper demethylation and expression.

The main shortcomings of the paper are the marginal differences in bone sizes in TKO model and lack of key pieces of evidence in support of the model that they propose. Mainly, they do not provide any evidence that loss of Tets affect Runx2 occupancy as they conclude in their study. They also don't show that Tet1/2/3 occupy similar sites or get recruited to similar sites in the absence of one another.

Some specific comments regarding experiments and figures are listed below:

Figure. 1: The differences in measurements and sizes of all bones are very marginal. Images don't appear to be on same scale.

Figure S1: The increase in osteogenesis in response to Aza treatment is marginal.

Figure 2: The loss of Tets, in particular Tet3, is partial suggesting mosaic deletion. No IF to show this.

Figure S2: The mouse model is not well validated. There is no test for expression of Tet1/2/3 outside of bones to ensure there is no non-specific deletion. This Cre line does have some expression in the germ line. Also, mice are likely infertile because of skeletal issues as well as germ cell defects. There is no documentation of mating etc... to support this, commenting on fertility in this paper, as they do, is not needed.

Figure 3: The data is only pertinent to Tet1 catalytic activity so should not be over concluded as catalytic activity of all Tets. Be specific.

Figure S3: Clearly Tet2 appears to have the most impact than Tet1 and Tet3 on differentiation so that should be emphasized.

Figure 4: No pursuit of upregulated genes. Nearly half of differentially expressed genes are upregulated and no comments or experiments to explain.

Figure 5: There is no Tet1/2/3 ChIP-seq to show how these proteins occupy their targets. In 5E, in text it says occupancy of Tets and Runx2 but that is not what the panel shows. Also, there are 277K DMRs and 40K DORs. There is 25% overlap of DORs to DMRs (is it significant considering the unbelievably large number of DMRs)

Figure 6: IPs are all in overexpression 293T cells and not in Ost or other bone related cells. Focus is all on Tet2 while conclusions are drawn on all three Tets. Also IgG controls are lacking in the IPs here and in Figure S8.

Figure 7: There is no ChIP-seq or ChIPqPCR to show that Tet2 is recruited to the Runx2 targets (perhaps some published data is provided). Not evidence that in the absence of Tet2 Runx2 occupancy is affected (extremely critical piece of evidence lacking for the model they propose). There is just some circumstantial evidence that methylation at Runx2 motifs or peaks is affected in TKO system.

Minor point: English writing is weak, some sentences don't read fluently or use of casual non-scientific terms like screenshot of data, bunch of genes etc... are used.

Anonymous review (not signed)

Reviewer #2 (Remarks to the Author):

Overall the manuscript by Wang et al. is a strong study examining the effect of Tet1-3 in skeletal tissue development. The study has a moderate novelty factor based on a previously published paper by Yang et al. Nat Commun paper, which examined Tet1 global KO/ Tet2 KO in bone. That study reported the importance of Tets in bone formation via Runx2 and exosomes. Also the paper by Cakouros et al. Epigen & Chromatin 2019 (should be referenced) showed that Tet1 and 2 but not Tet3 are important in human

MSC growth and fate determination where Tet1 and 2 affected Runx2 expression and 5hmC status. The present manuscript provides additional knowledge to the field by showing that Runx2 directly interacts with Tet1-3. The suggested redundancy in bone development is intriguing but is at odds with other studies showing that Tet3 seems to have no functional effect on human MSC and is lowly expressed by mouse MSC. This should be discussed further. Major issues are listed below.

1. Researchers should look at bone marrow fat in triple knockout mice.
2. Are any pathways involved in adipogenesis affected?
3. In the triple knockout, there are 22734 hypomethylated regions. How do you explain this given Tets remove DNA methylation.
3. bZIP, CTF, bHLH, and TEA/TEAD were identified as factors overlapping the DMR. Would be beneficial to show that they are functionally important in your system by knocking them down to show effects on differentiation.
4. ATAC seq identified two fold more OCR compared to COR. Wouldn't that result in active gene expression hence more genes will be activated? If there are more regions hypermethylated, why would that lead to more OCR than COR?
5. Line 267 says "also bound by Runx2" Should be "also bound by Runx2".
6. CHIP in Figure 7 is about PolII recruitment. Should also include 5hmC CHIP along this region.
7. WGBS is performed in this paper which is very informative however there is a need to do genome wide 5hmC to determine if this overlaps with regions that become hyper or hypomethylated. Tet could simply be inhibiting DNA methyltransferase from accessing DNA or hydroxymethylating DNA to remove the methyl mark. A lot of information can be obtained doing this experiment.

Reviewer #3 (Remarks to the Author):

Wang et al., generated the TCKO mice of Tet1, Tet2, and Tet3 using Prx1-Cre and showed the delayed calvarial development and shortening of limb bones and clavicles. In the overlapped region of hyper-CG-DMR and OCR in TCKO cells, they found that the binding motifs of Runt, CTF, bZIP, TEA/TEAD, and RHD were enriched, and Tet bound to Runx2 but not the other family gene proteins. They also showed that the peaks of Runx2 ChIP seq matched to DMR and the reduced peaks of ATAC in Runx2 target genes, including *Alpl*, *Col1a1*, and *Col1a2*, in TCKO primary osteoblasts. They indicated that the interaction of

Tet and Runx2 demethylates the DNA in Runx2 binding regions, activates Runx2 target gene expression, and induces bone development.

Major points:

1. This paper is potentially interesting. The most interesting part of this work is the phenotypes of TCKO mice. They showed the delayed calvarial development and shortening of limb bones and clavicles. They also showed the delay in trabecular bone formation at diaphysis of limb bones at E16.5 and the delay in bone formation in valvaria at P0 by histological analysis. The most striking phenotype of TCKO mice is severe dwarfism. In histological analysis of femurs at P0, bone marrow seemed to be occupied mostly by bone, indicating that bone formation is accelerated or bone resorption is inhibited. First, the chondrocyte differentiation, proliferation, and apoptosis need to be investigated to understand the reasons for dwarfism. In situ hybridization using the probes for chondrocyte marker genes, BrdU labeling, and TUNEL or caspase3 staining are required for them. In situ hybridization using osteoblast marker genes is also required for limb bones at different developmental stages. TRAP staining is also needed to examine the cartilage and bone resorption. Although the phenotypes of delayed calvarial development and shortened clavicles were emphasized, the phenotypes of TCKO mice are quite different from Runx2^{+/-} mice, which do not show apparent dwarfism. Thus, thorough analysis of the phenotypes of TCKO mice is essential before pursuing the molecular mechanism of the phenotypes.

2. Most of the peaks of ATAC seq in TCKO were similarly lower than those in WT, the different methylation did not specifically affect the peak level of ATAC seq, and the regions with reduced ATAC peaks were not specifically associated with those with Runx2 ChIP seq peaks in genome loci of *Alpl*, *Col1a1*, and *Col1a2*. Further, there are many peaks in Runx2 ChIP seq, and the physiologically important peaks are not clarified in most or all of the Runx2 target genes. Further, there are many Runx binding motifs in whole genome, most of which are not included in the regions of Runx2 ChIP seq peaks. Therefore, it is still insufficient to indicate that Tet activates Runx2 target gene expression by interacting with Runx2 and reducing the methylation of DNA, although they found the enrichment of Runx binding motifs in the overlapped region of hyper-CG-DMR and OCR and the interaction of Runx2 and Tet.

Minor points:

1. The three-week-old mice are not adult (page 5).

2. On page 12, it is difficult to indicate that the interaction between bone and neuron innervation was also regulated by TET enzymes.

3. In Fig. 1b and 1h, unmineralized area in calvaria should be quantitated.

4. Specify the name of bone used in Fig. 2c, S2c.
5. RT-qPCR using bone tissues are required in Fig. 2i.
6. Explain why the reductions in Tet^{HD}/fl were more severe than those in Tet^{1fl}/fl in Fig. 3h, i.
7. Describe Fag-TetFL and Flag-TetN-term in Fig. 6b in Methods.
8. Does it need to purify by anti-FLAG beads before incubation with GST (Fig. 6c)?
9. Does OE mean overexpression in Fig. 4d?
10. Origin of DMR should be shown in Fig. 7a.
11. What is TTS in Fig. S4g?
12. Abbreviation of POB for MC3T3-E1 cells is confusing in Fig. S6c-e.

REVIEWER COMMENTS

Reviewer #1 (Remarks to the Author):

In this study Zou and colleagues delete Tet enzymes in the mesenchymal stem cells using Prx1-Cre and find that it leads to defects in embryonic and adult bone development. They further show that deficiency of Tets leads to deregulation of osteogenic genes and increased DNA methylation, specifically at RUNX2 binding sites, in osteoblasts. They show that Tets interact with RUNX2 in 293T cells when overexpressed, and conclude that Tet enzymes facilitate RUNX2 recruitment to its target osteogenic genes for their proper demethylation and expression.

The main shortcomings of the paper are the marginal differences in bone sizes in TKO model and lack of key pieces of evidence in support of the model that they propose. Mainly, they do not provide any evidence that loss of Tets affect RUNX2 occupancy as they conclude in their study. They also don't show that Tet1/2/3 occupy similar sites or get recruited to similar sites in the absence of one another.

Thank the reviewer for raising these concerns. The bone size differences at P0 looks marginal. However, as Reviewer 3 suggested, 'the most interesting part of this work is the phenotypes of TCKO mice'. *Tet1*, *Tet2* and *Tet3* triple conditional knockout (TCKO) mice showed multiple cleidocranial dysplasia like skeletal defects, including delayed calvarial development and shortening of limb bones and clavicles. Additionally, the mineral deposition (Fig. 2c), and the expression of osteopontin and osteocalcin were decreased in the skull of TCKO mice (Fig. 2d and Supplementary Fig. 3a). TCKO mice also exhibited less mineralization in midshaft of the femur by Von-Kossa staining (Fig. 2e). These data supported that the loss of TETs affected bone development.

The reviewer also concerned that there is little evidence that prove TET affects RUNX2 occupancy. We re-analyzed the RUNX2 ChIP-seq data and showed that RUNX2 binding regions have strong correlation with the regions that displayed decreased CG methylation level, which were determined by WGBS assay. RUNX2 binding regions also have strong correlation with the opened chromatin regions, which were determined by ATAC-seq assay (Fig. 5d). More importantly, compared to the WT cells, TCKO cells displayed increased CG methylation level and decreased chromatin opening in the RUNX2 occupied region (Fig. 5d), supporting that TET1/2/3 occupy similar sites with RUNX2 or are recruited to similar sites by RUNX2. We also analyzed some RUNX2 downstream genes, including *Alpl*, *Col1a1* and *Col1a2*. As shown in supplementary Fig. 9c-9e, TETs deficiency caused increased 5mc levels and decreased chromatin opening in the RUNX2 occupied region. During the revision, we also did RUNX2 ChIP analysis using osteoblasts from WT and TCKO mice. As expected, we found that RUNX2 occupancy in the promoter of its downstream genes were significantly decreased in the TCKO cells, supporting that

loss of TETs impaired the RUNX2 occupancy (Fig. 7e-7g). And in the promoter regions of these genes, the enrichments of RNA polymerase II are also reduced (Fig. 7h-7j). These data support that loss of TETs can affect the occupancy of RUNX2 and the transcription of its target genes.

Figure 5. (d). Heatmap and metaplot of RUNX2 ChIP-seq , CG methylation level, and ATAC-seq signals of TCKO and WT cells over RUNX2 binding sites.

Supplemental Figure 9. (c-e) Screenshot of indicated genes from RUNX2 ChIP-seq and input in pre-osteoblastic MC3T3-E1 cells (pre-osteoblast, POB) (GSM1027478), CG methylation level from WGBS, ATAC-seq, 5hmC-Seal, RNA-seq in WT and TCKO primary osteoblasts over osteogenic marker genes including *Alpl* (c), *Col1a1* (d), and *Col1a2* (e) genes. Number with the brackets indicate the relative levels for ChIP-seq, WGBS, ATAC-seq and RNA-seq data.

Figure 7. (e-g) ChIP-qPCR for RUNX2 on the promoter regions of osteogenic genes including *Col1a1* (e), *Col1a2* (f), *Alpl* (g). **(h-j)** ChIP-qPCR for RNA Polymerase II (RNA Pol II) on the promoter regions of osteogenic genes including *Col1a1* (h), *Col1a2* (i), *Alpl* (j).

The reviewer also raised a concern of lacking evidence that TET1/2/3 occupy similar sites. We agree with the reviewer that it is worthy to further determine the fine-tune regulation by different TET proteins during osteoblast differentiation and bone development. We have tried to perform ChIP-assay using different TET1/2/3 antibodies to compare their different binding regions. Unfortunately, the antibodies did not work well in the ChIP assay and it is hard to directly compare the binding differences of TETs proteins. Considering that the bone defects in TCKO mice can be rescued by the presence of any single allele of *Tets* (Fig. 1b-1g) and RUNX2 can interact with all of the three TETs and their catalytic domain (Fig. 6a-6b), we inferred that three TETs could bind some similar regions. We provided more *in vitro* data to confirm the interaction between RUNX2 and different TET proteins in revised manuscript.

Figure 6. (a) Co-immunoprecipitation (Co-IP) of Flag-TET1 or Flag-TET2 or Flag-TET3 with HA-RUNX2. HA-RUNX2 expressing plasmid was co-transfected with Flag-TET1 or Flag-TET2 or Flag-TET3 in 293T cells. Whole cell lysate was used for immunoprecipitation and then blotted with indicated antibodies. **(b)** Co-immunoprecipitation (Co-IP) of Flag-TET1-CD or Flag-TET2-CD or Flag-TET3-CD with HA-RUNX2. HA-RUNX2 expressing plasmid was co-transfected with Flag-TET1-CD or Flag-TET2-CD or Flag-TET3-CD in 293T cells. Whole cell lysate was used for immune-precipitation and then immunoblotting with indicated antibodies.

Some specific comments regarding experiments and figures are listed below:

Figure. 1: The differences in measurements and sizes of all bones are very marginal.

Images don't appear to be on same scale.

Thank the reviewer for raising this concern. In terms of skeletal development, TCKO mice displayed typical skeletal defects of the cleidocranial dysplasia at P0, including short limbs, delayed closure of cranial sutures and clavicle hypoplasia, etc. We added the histological staining images of limbs to further show the differences between wild type mice and TCKO mice (Supplementary Fig. 3c). Following the reviewer's suggestions, we have cut the images again from the original pictures to make the sizes consistent in the revised Fig 1a-1b and Fig 3c.

Supplemental Figure 3. (c) Safranin O staining of limbs isolated from WT and TCKO mice at different developmental stages. Scale bar = 1mm.

Figure S1: The increase in osteogenesis in response to Aza treatment is marginal. Thank the reviewer for raising this concern. We agree with the reviewer that the effects of 5-Aza treatment on the osteogenesis is marginal. And we repeated the experiments, and found that 5-Aza treatment could consistently increase the osteogenesis (Figure R1).

Figure R1. ALP activity quantification was measured by phosphatase substrate assay as A405/Ala.Blue at day7 of the differentiation. * $P < 0.05$, ** $P < 0.01$. Two-tailed Student's t-test. Data are presented as mean \pm s.d., $n = 4$.

Figure 2: The loss of Tets, in particular Tet3, is partial suggesting mosaic deletion. No IF to show this.

Thank the reviewer for this question. Considering osteoblasts isolated from calvarial

bone may be contaminated by some non-targeted cells, we repeated the experiments and optimized the osteoblast isolation protocol by cleaning calvarial bones to reduce the contamination. As shown in the revised Fig. 2i, the knock out efficiency was substantially increased. We replaced the data in the revised manuscript.

Figure 2. (i) RT-qPCR analysis of *Tet1*, *Tet2*, *Tet3*, *Sp7*, *Alp*, *Col1 α 1* and *Bglap* expression after osteoblast differentiation for 7 days, cells were from WT and TCKO mice. * $P < 0.05$, ** $P < 0.01$. Two-tailed Student's t-test. Data are presented as mean \pm s.d., $n = 4$.

Figure S2: The mouse model is not well validated. There is no test for expression of *Tet1/2/3* outside of bones to ensure there is no non-specific deletion. This Cre line does have some expression in the germ line. Also, mice are likely infertile because of skeletal issues as well as germ cell defects. There is no documentation of mating etc... to support this, commenting on fertility in this paper, as they do, is not needed.

Thank the reviewer for these suggestions. As the reviewer suggested, we examined the expression of *Tet1/2/3* in the liver and revealed there is no non-specific deletion in the liver of *Prx1-Cre* TCKO mice (Supplementary Fig. 2a-2b). The reviewer pointed out that *Prx1-Cre* line does have some expression in the germ line. Actually, we have noticed that *Prx1-Cre* mouse line could be expressed transiently in the oocytes, thus we designed the breeding strategy by using male *Prx1 cre*, for example *Prx1 cre*; *Tet1^{fl/fl} Tet2^{fl/fl} Tet3^{fl/+}* mice with female *Tet1^{fl/fl} Tet2^{fl/fl} Tet3^{fl/fl}* mice to avoid the *Tets* deletion in the oocytes. We have added some information in terms of breeding strategy into the methods. We have removed the discussion on fertility as the reviewer suggested.

Supplemental Figure 2. (a-b) RT-qPCR analysis of *Tet1*, *Tet2* and *Tet3* expression in cranial

bones (**a, left**) and femurs (**a, right**), and liver (**b**) of WT and TCKO mice. * $P < 0.05$, ** $P < 0.01$. Two-tailed Student's t-test. Data are presented as mean \pm s.d., cranial bone: $n = 3$; liver: $n = 2$.

Figure 3: The data is only pertinent to Tet1 catalytic activity so should not be over concluded as catalytic activity of all Tets. Be specific.

Thank the reviewer for this kind suggestion. We have restrained the description to the TET1 mice to avoid the over-conclusion in the revised manuscript.

Figure S3: Clearly Tet2 appears to have the most impact than Tet1 and Tet3 on differentiation so that should be emphasized.

Thank the reviewer for this suggestion. We have emphasized this point in revised MS as "Among the three CD domains, the TET2-CD has the most impact than TET1-CD and TET3-CD on calcium nodule formation in late stage of osteogenic differentiation (Fig. 3a bottom)."

Figure 4: No pursuit of upregulated genes. Nearly half of differentially expressed genes are upregulated and no comments or experiments to explain.

Thank the reviewer for this suggestion. Gene ontology (GO) enrichment analysis of these different expression genes (DEGs) indicated that bone morphogenesis and ossification were top enriched GO terms for down-regulated genes in TCKO cells, while axonogenesis related GO terms was enriched in up-regulated genes (Supplementary Fig. 5b). Considering the bone defects in the TCKO mice, we focused on the bone morphogenesis and ossification associated genes in this paper. As the reviewer suggested, we made a discussion in the revised MS as follow:

"We identified 432 down-regulated and 431 up-regulated differential expressed genes (fold change greater than 2 and FDR less than 0.05) in the TCKO osteoblasts. Overwhelming evidences support that 5mC has a negative effect on transcription. However, intragenic DNA methylation is highest in transcriptionally active genes, where it is reported to influence both elongation efficiency and splicing¹. The up-regulated genes may be activated by specific increased intragenic DNA methylation. Otherwise, these genes could be a secondary effect of TET proteins deletion. The specific reasons need further analysis and experiments. In the up-regulated genes, the neuroactive ligand-receptor interaction and axon guidance pathway were enriched in TCKO cells. In the up-regulated genes, osteoblast derived *Cox2* has been reported to promote the innervation of sensory nerve fibers². The sensory nerves can sustain the bone mass accrual³. This could be a compensated result of bone loss in TCKO mice."

Figure 5: There is no Tet1/2/3 ChIP-seq to show how these proteins occupy their targets. In 5E, in text it says occupancy of Tets and RUNX2 but that is not what the panel shows. Also, there are 277K DMRs and 40K DORs. There is 25% overlap of

DORs to DMRs (is it significant considering the unbelievably large number of DMRs)

We thank the reviewer for this suggestion. We have tried to perform ChIP assay of TET1/2/3 in the absence of one another, but the three antibodies did not work well. We reasoned that hyper-CG-DMR in TCKO cells reflected the binding sites of TET1/2/3. In addition, to address reviewer's concern, we also performed 5hmC-Seal⁴ in TCKO and WT osteoblasts to map genome-wide 5hmC signals. Our new results showed significant changes of 5hmC signals with hyper-CG-DMRs in TCKO cells (Fig. 5a).

To avoid confusion, instead of referring to "regions bound by TET1/2/3", we now revised it as "hyper-CG-DMRs" following the reviewer's suggestions. As shown in the following sentence: "We observed strong enrichment of down regulated genes with overlapped regions of Runx2 binding sites and **hyper-CG-DMRs**, especially if those regions appeared in their promoters and potential enhancers (Fig. 5e). GO term analysis of those downregulated genes indicated ossification as the most enriched terms, suggesting those genes could be the potential targets of RUNX2 and TET1/2/3 (Fig. 5f)."

For the concern of a large number of DMRs, we re-analyzed DMRs with different CG methylation differences cut-off. In parallel, we also investigated the number of overlapped regions for ATAC-seq oC-Rs to all those hyper-CG-DMRs (Supplementary Fig. 5g). Based on our new results, we found that methylation difference cut-off (0.2, 0.3, 0.4, 0.5, 0.6, 0.7, 0.8, 0.9) resulted in different number of hyper/hypo-CG-DMRs called, however, the number of hyper-CG-DMRs overlapped with oC-Rs is similar (around 25%). Our new analysis indicated that the number of hyper-CG-DMRs overlapped with oC-Rs became saturated when we used 0.4 as the methylation difference cut-off. Thus, we applied this cut-off for DMR analysis in the revised manuscript. With this cut-off, we identified 86598 hyper-CG-DMRs, 1944 hypo-CG-DMRs and 10030 overlapped regions of hyper-CG-DMRs and oC-Rs in TCKO condition (Fig. 4f and Fig. 5b). All our downstream analysis has been repeated using this new cut-off and the results we obtained support our initial conclusions.

Figure 5. (a) Association analysis of RNA-seq, WGBS, 5hmC-Seal, ATAC-seq, and

RUNX2 ChIP-seq data suggest TET and RUNX2 proteins could synergistically promote osteogenesis.

g

Supplemental Figure 5. (g) Number of hyper-CG-DMR, hypo-CG-DMR, and overlap region of hyper-CG-DMR and chromatin Open-to-Close-Region (OCR) with different thresholds for calling DMR.

f

Figure 4. (f) Pie chart of hypermethylated and hypomethylated CG DMR counts in TCKO and WT cells.

Figure 5. (b) Venn diagram of hyper-CG-DMR and chromatin Open-to-Close-Region (oCR) in TCKO cells.

Figure 6: IPs are all in overexpression 293T cells and not in Ost or other bone related cells. Focus is all on Tet2 while conclusions are drawn on all three Tets. Also IgG controls are lacking in the IPs here and in Figure S8.

Thank the reviewer for these suggestions. As the reviewer suggested, we tried to perform the IPs using C3H10T1/2 cells. As we have not got effective antibodies of TETs for immunoprecipitation, we infected the C3H10T1/2 cells with the lentivirus expressing Flag tag-catalytic domain of TET1 and TET2 and pulled down TET1/2 catalytic domain with Flag beads. And then we did immunoblot using anti-RUNX2 antibody. As shown in Supplementary Fig. 12c, we could see the interaction between TET1/2 catalytic domain and endogenous RUNX2 in C3H10T1/2 cells. We also added some evidence that RUNX2 can interact with TET1/2/3 catalytic domain (Fig. 6b).

In our Co-IP experiments here, we used anti-Flag beads, instead of IgG antibody as the control to rule out non-specific binding. In the absence of Flag tagged protein, anti-Flag beads could not pull HA-tagged RUNX2 proteins down. In our Co-IP experiments in the original Figure S8, we used anti-HA beads. In the same way, anti-HA beads could not pull Flag-tagged proteins down in the absence of HA tagged protein. We provided some information into the methods in the revised line 454-458.

Supplemental Figure 12. (c) Co-immunoprecipitation (Co-IP) of Flag-TET1-CD or Flag-TET2-CD with endogenous RUNX2 in C3H10T1/2 cells. Flag-TET1-CD or Flag-TET2-CD lentivirus was infected the C3H10T1/2 cells. Whole cell lysate was used for immunoprecipitation with Flag-Beads and then blotted with indicated antibodies.

Figure 6. (b) Co-immunoprecipitation (Co-IP) of Flag-TET1-CD or Flag-TET2-CD or Flag-TET3-CD with HA-RUNX2. HA-RUNX2 expressing plasmid was co-transfected with Flag-TET1-CD or Flag-TET2-CD or Flag-TET3-CD in 293T cells. Whole cell lysate was used for immune-precipitation and then immunoblotting with indicated antibodies.

Figure 7: There is no ChIP-seq or ChIPqPCR to show that Tet2 is recruited to the RUNX2 targets (perhaps some published data is provided). Not evidence that in the

absence of Tet2 RUNX2 occupancy is affected (extremely critical piece of evidence lacking for the model they propose). There is just some circumstantial evidence that methylation at RUNX2 motifs or peaks is affected in TKO system.

Thank the reviewer for raising this concern. As we have not got effective antibodies of TETs for immunoprecipitation, we used the 5mc and 5hmc as an indirect manner to determine the potential binding regions of TETs. By this way, we found that RUNX2 binding sites are similar with these potential regions. Additionally, we found that loss of TETs decreased the RUNX2 binding in the promoter regions of its target genes (Fig. 7e-7g). And in the promoter regions of these genes, the enrichments of RNA polymerase II are also reduced (Fig. 7h-7j). These data support that loss of TETs can affect the occupancy of RUNX2 and the transcription of its target genes.

Figure 7. (e-g) ChIP-qPCR for RUNX2 over the promoter regions of osteogenic genes including *Col1a1* (e), *Col1a2* (f), *Alpl* (g). (h-j) ChIP-qPCR for RNA Polymerase II (RNA Pol II) over the promoter regions of osteogenic genes including *Col1a1* (h), *Col1a2* (i), *Alpl* (j).

Minor point: English writing is weak, some sentences don't read fluently or use of casual non-scientific terms like screenshot of data, bunch of genes etc... are used.

Thank the reviewer for this suggestion. We have checked the article carefully. And highlight the changes.

Anonymous review (not signed)

Reviewer #2 (Remarks to the Author):

Overall the manuscript by Wang et al. is a strong study examining the effect of Tet1-3 in skeletal tissue development. The study has a moderate novelty factor based on a previously published paper by Yang et al. Nat Commun paper, which examined Tet1 global KO/ Tet2 KO in bone. That study reported the importance of Tets in bone formation via RUNX2 and exosomes. Also the paper by Cakouros et al. Epigen & Chromatin 2019 (should be referenced) showed that Tet1 and 2 but not Tet3 are important in human MSC growth and fate determination where Tet1 and 2 affected RUNX2 expression and 5hmC status. The present manuscript provides additional knowledge to the field by showing that RUNX2 directly interacts with Tet1-3. The suggested redundancy in bone development is intriguing but is at odds with other studies showing that Tet3 seems to have no functional affect on human MSC and is

lowly expressed by mouse MSC. This should be discussed further. Major issues are listed below.

1. Researchers should look at bone marrow fat in triple knockout mice.

Thank the reviewer for this suggestion. We found that there is no difference of marrow fat before 2 weeks between WT and TCKO mice. But, the marrow fat accumulation was observed in TCKO mice at 9 weeks (Figure R2).

Figure R2. (a-b) Immunofluorescence of adipocyte marker Perilipin in bone marrow from WT and TCKO mice at 2-week and 9-week. Scale bar = 200µm.

2. Are any pathways involved in adipogenesis affected?

Thank the reviewer for this suggestion. We have checked GO terms of both up and down regulated genes, but no adipogenesis related GO terms were found. The possible reason of these phenomena might be that we used osteoblasts for RNA sequence.

3. In the triple knockout, there are 22734 hypomethylated regions. How do you explain this given Tets remove DNA methylation?

We thank the reviewer for this comment. As mentioned by the reviewer, we should only capture hyper-CG-DMRs in TCKO as TETs remove DNA methylation. However, as we are calculating DMRs based on some statistical models, there could always be some technical noise. In our previous version of manuscript, we applied a relative lose cut-off to define the DMR (0.2 methylation differences for 200bp bins and FDR less than 0.05). With different methylation cut-offs (0.2, 0.3, 0.4, 0.5, 0.6, 0.7, 0.8, 0.9), the number of DMRs is different. Given DNA methylation is anticorrelate with chromatin accessibility, we also analyzed the number of overlapped regions of DMRs with ATAC-seq oC-Rs in TCKOs and we observed that the overlap become saturated using 0.4 as the methylation difference cut-off. Thus, we applied this cut-off for DMR analysis in the revised manuscript. With this cut-off, we identified 86598 hyper-CG-DMRs, and 1944 hypo-CG-DMRs which we believe to be background noises (Supplemental Fig.4f and Supplemental Fig.5b). All our downstream analysis has been repeated using this new cut-off and the results we obtained support our initial conclusions.

Supplemental Figure 5. (g) Number of hyper-CG-DMR, hypo-CG-DMR, and overlapped region of hyper-CG-DMR and chromatin Open-to-Close-Region (oC-R) with different thresholds for calling DMR.

Figure 4. (f) Pie chart of hypermethylated and hypomethylated CG DMR counts in TCKO and WT cells.

3. bZIP, CTF, bHLH, and TEA/TEAD were identified as factors overlapping the DMR. Would be beneficial to show that they are functionally important in your system by knocking them down to show effects on differentiation.

Thank the reviewer for this suggestion. *Tead1-4* has been reported to inhibits osteogenesis before⁵. Thus, we performed the knock down assay with *Atf3*, *Nf1* and *Pitx1* to determine their effects on osteogenesis. And we found that loss of *Atf3* and *Nf1* can inhibit osteogenesis determined by ALP activity and osteogenic markers. Additionally, *Pitx1* showed little effects on osteoblast differentiation (Supplementary Fig. 7).

Supplemental Figure 7. (a) ALP staining and Alizarin red S staining after osteoblast differentiation for 7 days (top) and 21 days (bottom), respectively. Scale bar = 1mm. (b) ALP activity quantification was measured by phosphatase substrate assay as A405/Ala.Blue. * $P < 0.05$. Two-tailed Student's t-test. Data are presented as mean \pm s.d., $n=4$. (c-d) RT-qPCR analysis of indicated gene expression after osteoblast differentiation for 7 days. * $P < 0.05$, ** $P < 0.01$. Two-tailed Student's t-test. Data are presented as mean \pm s.d., $n = 4$.

4. ATAC seq identified two fold more OCR compared to COR. Wouldn't that result in active gene expression hence more genes will be activated? If there are more regions hypermethylated, why would that lead to more OCR than COR?

Thank you for this comment. OCR (Open-to-Close-Region) meant that chromatin states shifting from open to close in our manuscript. So the result of ATAC-seq showed the same tendency with RNA-seq and WGBS. The deletion of TET1/2/3 proteins in TCKO mice resulted in more hypermethylated regions and more closed regions in a genome-wide scale. We realized using OCR or COR might be confusing. In this revised version, we change it to oC-R (refer to Open-to-Close-Region) and cO-R (refer to Close-to-Open-Region) for clarity.

5. Line 267 says "also bond by RUNX2" Should be "also bound by RUNX2".

Thank the reviewer for pointing this spelling error, and we corrected it in the revised MS line 307.

6. ChIP in Figure 7 is about PolIII recruitment. Should also include 5hmc ChIP along this region.

Thank the reviewer for this suggestion. Following the suggestions from the reviewer, we performed 5hmC-Seal to characterize the genome wide 5hmC patterns. We added new data about the 5hmC-Seal tracks into the revised Figures (Supplementary Fig. 9c-9e).

Supplemental Figure 9. (c-e) Screenshot of indicated genes from RUNX2 ChIP-seq and input in pre-osteoblastic MC3T3-E1 cells (pre-osteoblast, POB) (GSM1027478), CG methylation level from WGBS, ATAC-seq, 5hmC-Seal, RNA-seq in WT and TCKO primary osteoblasts over osteogenic marker genes including *Alpl* (c), *Col1a1* (d), and *Col1a2* (e) genes. Number with the brackets indicate the relative levels for ChIP-seq, WGBS, ATAC-seq and RNA-seq data.

7. WGBS is performed in this paper which is very informative however there is a need to do genome wide 5hmC to determine if this overlaps with regions that become hyper or hypomethylated. Tet could simply be inhibiting DNA methyltransferase from accessing DNA or hydroxymethylating DNA to remove the methyl mark. A lot of information can be obtained doing this experiment.

We thank the reviewer for this suggestion. We have performed the genome wide 5hmC sequencing. Based on the integrated analysis of WGBS, 5hmC-Seal, and ATAC-seq data, we found that a significant down regulated signal of 5hmC and ATAC-seq in hyper-CG-DMRs in TCKO cells when compared to WT cells (Fig.5a). The result is consistent with previous studies that TET proteins catalyze the successive oxidation of 5-methylcytosine (5mC) to 5-hydroxymethylcytosine (5hmC), 5-formylcytosine (5fC), and 5-carboxylcytosine (5caC) to increase chromatin accessibility.

Figure 5. (a) Heatmap and metaplot of CG methylation level, 5hmC-Seal and ATAC-seq signals over hyper-CG-DMR in WT and TCKO cells.

Reviewer #3 (Remarks to the Author):

Wang et al., generated the TCKO mice of Tet1, Tet2, and Tet3 using Prx1-Cre and showed the delayed calvarial development and shortening of limb bones and clavicles. In the overlapped region of hyper-CG-DMR and OCR in TCKO cells, they found that the binding motifs of Runt, CTF, bZIP, TEA/TEAD, and RHD were enriched, and Tet bound to RUNX2 but not the other family gene proteins. They also showed that the peaks of RUNX2 ChIP seq matched to DMR and the reduced peaks of ATAC in RUNX2 target genes, including *Alpl*, *Col1a1*, and *Col1a2*, in TCKO primary osteoblasts. They indicated that the interaction of Tet and RUNX2 demethylates the DNA in RUNX2 binding regions, activates RUNX2 target gene expression, and induces bone development.

Major points:

1. This paper is potentially interesting. The most interesting part of this work is the phenotypes of TCKO mice. They showed the delayed calvarial development and shortening of limb bones and clavicles. They also showed the delay in trabecular bone formation at diaphysis of limb bones at E16.5 and the delay in bone formation in valvaria at P0 by histological analysis. The most striking phenotype of TCKO mice is severe dwarfism. In histological analysis of femurs at P0, bone marrow seemed to be occupied mostly by bone, indicating that bone formation is accelerated or bone resorption is inhibited. First, the chondrocyte differentiation, proliferation, and apoptosis need to be investigated to understand the reasons for dwarfism. In situ hybridization using the probes for chondrocyte marker genes, BrdU labeling, and TUNEL or caspase3 staining are required for them. In situ hybridization using

osteoblast marker genes is also required for limb bones at different developmental stages.

TRAP staining is also needed to examine the cartilage and bone resorption. Although the phenotypes of delayed calvarial development and shortened clavicles were emphasized, the phenotypes of TCKO mice are quite different from RUNX2^{+/-} mice, which do not show apparent dwarfism. Thus, thorough analysis of the phenotypes of TCKO mice is essential before pursuing the molecular mechanism of the phenotypes.

Thank the reviewer for these suggestions. The reviewer very astutely points out that the bone occupied the bone marrow at P0. Following the reviewer's suggestions, we firstly examined chondrocyte differentiation in vitro. We found that the chondrocyte differentiation is impaired in TCKO cells, compared with wild type chondrocytes (Supplementary Fig. 3d-3e). Then we examined chondrocyte in vivo and observed the aberrant chondrocyte proliferation in resting zone in TCKO mice, determined by PCNA staining and Edu assay (Supplementary Fig. 3f). The occupied bone at the bone marrow at P0 could be from the increased number of chondrocytes. We also performed the TRAP assay as the reviewer suggested. And the osteoclasts were not affected at P0 (Supplementary Fig. 4a), suggesting that the occupied bone is not from the dysfunction of osteoclast. Besides, we also observed the increase of chondrocyte apoptosis in the proliferation zone in TCKO mice (Supplementary Fig. 3g), suggesting the impaired endochondral ossification process. The dwarfism in TCKO mice, which is more severe at 3-week (Supplementary Fig. 2d), could be from this impaired endochondral ossification process. We discussed this dwarfism phenotype in the revised paper. In our current study, we focused on the osteogenesis and found the repressed osteogenesis (Fig. 2j-2i) in TCKO mice. Additionally, we added in situ hybridization results of the osteoblast marker gene *Col1* at different developmental stages (Supplementary Fig. 3b) as the reviewer suggested.

Supplemental Figure 3. (b) RNA in situ hybridization of *Col1a1* of the femur from WT and TCKO mice at E15.5 and E16.5. Scale bar = 500 μ m. (c) Safranin O staining of limbs isolated from WT and TCKO mice at different developmental stages. Scale bar = 1mm. (d) Alcian blue staining of chondrocyte differentiation at day 7. (e) RT-qPCR analysis of indicated genes expression after chondrocyte differentiation at day 7. * $P < 0.05$, ** $P < 0.01$. Two-tailed Student's t-test. Data are presented as mean \pm s.d., n = 4. (f) Immunofluorescence of proliferation marker PCNA and Edu in the femurs of WT and TCKO mice at P0. Scale bar = 500 μ m. (g) Immunofluorescence of cleaved-caspase 3 in the femurs of WT and TCKO mice at P0. Scale bar: 500 μ m for the top images and 40 μ m for the bottom images.

Supplemental Figure 4. (a) TRAP staining in the femurs of WT and TCKO mice at P0.

2. Most of the peaks of ATAC seq in TCKO were similarly lower than those in WT, the different methylation did not specifically affect the peak level of ATAC seq, and the regions with reduced ATAC peaks were not specifically associated with those with RUNX2 ChIP seq peaks in genome loci of *Alpl*, *Col1a1*, and *Col1a2*. Further, there are many peaks in RUNX2 ChIP seq, and the physiologically important peaks are not clarified in most or all of the RUNX2 target genes. Further, there are many Runx binding motifs in whole genome, most of which are not included in the regions of RUNX2 ChIP seq peaks. Therefore, it is still insufficient to indicate that Tet activates RUNX2 target gene expression by interacting with RUNX2 and reducing the methylation of DNA, although they found the enrichment of RUNX2 binding motifs in the overlapped region of hyper-CG-DMR and OCR and the interaction of RUNX2 and Tet.

Thank the reviewer for these comments. To address reviewer's concern, we added a new analysis for the number of Chromatin Open-to-Close-Region (oC-R, n=39643, 15.94%), Close-to-Open-Region (cO-R, n=13302, 5.20%), and Non-Difference-Region (ND-R, n=202019, 79.31%). As shown in Supplemental Fig.6e, majority of ATAC-seq peak regions in TCKO and WT show no significant differences, while only around 20% of ATAC-seq peak regions displayed differences.

Considering that the TET proteins can catalyze the oxidation of 5mC into 5hmC, we also examined the genome wide 5hmC levels in WT and TCKO osteoblasts using 5hmC-Seal⁴. To further reveal the relationship between the chromatin state change and TET related DNA demethylation, we plotted the WGBS, 5hmC-Seal, and ATAC-seq signals over oC-Rs and cO-Rs. As shown in Supplemental Fig.7, we observed stronger enrichment of differential CG methylation and 5hmC level in oC-R when compared to cO-Rs. These results support that TET related CG demethylation promote bone formation through the alteration of chromatin state. In addition, previous study showed that the 5hmC could promote transcriptional de-repression by dissociation of 5mC-binding proteins and/or recruitment of effector proteins⁶. As the 5hmC-Seal sequencing showed that loss of Tets gave rise to decreased 5hmC peaks, which is more related with the decreased chromatin accessibility in the genome loci of *Alpl*, *Col1a1*, and *Col1a2* (Supplementary Fig. 9c-9e).

We agreed with the reviewer that not all the reduced ATAC peaks were specifically associated with RUNX2 ChIP-seq peaks in the genome loci of *Alpl*, *Col1a1*, and *Col1a2*. But most of the RUNX2 ChIP-seq peaks were associated with decreased ATAC-seq peaks in these genes. As the reviewer suggested that there are many RUNX2 binding motif in the genome that was immunoprecipitated in this RUNX2 ChIP-seq, we performed the genome wide motif analysis in the overlap regions of hyper-CG-DMR and chromatin OCR in TCKO cells and found that RUNX2 binding motif was the top one in enriched known motifs (Fig. 5c), implicating that RUNX2 could associate with Tets to regulate gene expression. We have tried to perform ChIP-assay using TET1/2/3 antibody in the RUNX2 KO osteoblasts. Unfortunately,

the three antibodies did not work well in the ChIP assay.

Additionally, we performed ChIP-qPCR to determine the effects of loss of TETs in RUNX2 binding and transcription. We confirmed that RUNX2 could bind the *Col1a1*, *Col1a2*, *Alpl* genes (Fig. 7e-7g). And the RUNX2 enrichments were decreased in the promoter regions of its target genes (Fig. 7e-7g). And in the promoter regions of these genes, the enrichments of RNA polymerase II are also reduced (Fig. 7h-7j). These data support that loss of TETs can in turn affect the occupancy of RUNX2 and the transcription activity of its target genes.

Supplemental Figure 6. (e) Pie chart of chromatin Open-to-Close-Region (oC-R), Close-to-Open-Region (cO-R) and Non-Difference-Region (ND-R) in TCKO cells.

Supplemental Figure 8. (a-b) Heatmap and metaplot of CG methylation level, 5hmC-

Seal and ATAC-seq signals over chromatin Close-to-Open-Region (cO-R) (a) and Open-to-Close-Region (oC-R) (b) in TCKO cells.

Figure 7. (e-g) ChIP-qPCR for RUNX2 over the promoter regions of osteogenic genes including *Col1a1* (e), *Col1a2* (f), *Alpl* (g). **(h-j)** ChIP-qPCR for RNA Polymerase II (RNA Pol II) over the promoter regions of osteogenic genes including *Col1a1* (h), *Col1a2* (i), *Alpl* (j).

Minor points:

1. The three-week-old mice are not adult (page 5).

Thank the reviewer for this question. We corrected the sentence as "The TCKO mice were extremely short-limbed and weak at 3-week (Supplementary Fig. 2c)."

2. On page 12, it is difficult to indicate that the interaction between bone and neuron innervation was also regulated by TET enzymes.

Thank the reviewer for raising this concern. The genes associated with neuron innervation were up regulated in the TCKO osteoblasts. This could be a secondary effect of *Tets* deficiency. We also made some discussion in the revised MS.

3. In Fig. 1b and 1h, unmineralized area in calvaria should be quantitated.

Thank the reviewer for this suggestion. We annotated the unmineralized area under the image of calvarial bones in the revised Fig. 1b and made a statistically quantification of the μ -CT 3D images in the revised Fig. 1i

Figure 1. (b) The whole mount skeleton staining of WT and indicated genotype mice by Alcian blue and Alizarin red S at postnatal day 0 (P0). The quantification data of the unmineralized area was annotated below the cranial bones. scale bar = 5mm. (h) 3D μ -CT images of cranial bones isolated from indicated genotype mice at postnatal day 10 (P10). Representative images are from 2 independent samples. (i) Quantification of the unmineralized area of cranial bones in (h). $*P < 0.05$. Two-tailed Student's t-test. Data are presented as mean \pm s.d., $n = 2$.

4. Specify the name of bone used in Fig. 2c, S2c.

Thank the reviewer for this suggestion. The bones used in both Figure 2c and S2c are femurs, and we annotated this in the figure legend.

5. RT-qPCR using bone tissues are required in Fig. 2i.

Thank the reviewer for this question. We performed the RT-qPCR using the bone tissues including calvarial bones and femurs (Supplementary Fig. 2a-2b).

Supplemental Figure 2. (a) RT-qPCR analysis of *Tet1*, *Tet2* and *Tet3* expression in cranial bones (a, left) and femurs (a, right) of WT and TCKO mice. $*P < 0.05$, $**P < 0.01$. Two-tailed Student's t-test. Data are presented as mean \pm s.d., cranial bone: $n = 3$; femur: $n = 2$.

6. Explain why the reductions in TetHD/fl were more severe than those in Tet1fl/fl in Fig. 3h, i.

Thank the reviewer for this suggestion. We discussed the reason in the revised MS as "The osteogenic defects of osteoblasts isolated from *Prx1-Cre*, *Tet1^{fl/HD}* *Tet2^{fl/fl}* *Tet3^{fl/fl}* mice was more severe than that from TCKO mice (Figure 3i-3k). It is possible that early inactivation of TET1 from germ cells augmented the effects of TET1 loss in the skeletal lineage cells."

7. Describe Flag-TetFL and Flag-TetN-term in Fig. 6b in Methods.

Thank the reviewer for this suggestion. We described the information of plasmids

used in this paper as “Flag-TET1: full length; Flag-TET2: full length; Flag-TET3: full length; Flag-TET1-CD:1367aa-2008aa; Flag-TET2-CD, 1042aa-1913aa; Flag-TET3-CD:697aa-1669aa; Flag-TET1-Nterm,1aa-1366aa; Flag-TET2-Nterm, 1aa-1041aa; Flag-TET3-Nterm:1aa-696aa.” in the revised Methods.

8. Does it need to purify by anti-FLAG beads before incubation with GST (Fig. 6c)?

Thank the reviewer for this suggestion. We purified Flag-TET2-CD protein with the anti-Flag beads to concentrate the Flag-TET2-CD protein. The direct interaction between Flag-TET2-CD and GST-RUNX2 was done by using the two purified proteins.

9. Does OE mean overexpression in Fig. 4d?

Thank the reviewer for this suggestion. In Fig. 4d, OE mean overexpression. We deleted the “OE” to make it clear.

10. Origin of DMR should be shown in Fig. 7a.

Thank the reviewer for this suggestion. We annotated the sequence information of DMR in the method of luciferase reporter assay. Additionally, we made a diagram to make it clear in the revised Fig.7a.

Figure 7. (a) The strategy for promoter luciferase assay. OSE2: Osteocalcin-specific element 2. DMR: -2327bp to -1702bp of *Col1a1* promoter.

11. What is TTS in Fig. S4g?

Thank the reviewer for this question. TTS means transcription termination site. And we added the full name of UTR, ncRNA, and TTS in revised S5i.

12. Abbreviation of POB for MC3T3-E1 cells is confusing in Fig. S6c-e.

Thank the reviewer for this suggestion. According to the information of ATCC (American Type Culture Collection), MC3T3-E1 is a pre-osteoblast cell line that was originally isolated from new born mouse calvaria. And to avoid the confusion, we noted it in the figure legend as “MC3T3-E1 (pre-osteoblast, POB)”.

1. Greenberg, M.V.C. & Bourc'his, D. The diverse roles of DNA methylation in mammalian development and disease. *Nat Rev Mol Cell Biol* **20**, 590-607 (2019).
2. Chen, H. *et al.* Prostaglandin E2 mediates sensory nerve regulation of bone homeostasis. *Nat Commun* **10**, 181 (2019).
3. Fukuda, T. *et al.* Sema3A regulates bone-mass accrual through sensory innervations. *Nature* **497**, 490-3 (2013).

4. Song, C.X. *et al.* Selective chemical labeling reveals the genome-wide distribution of 5-hydroxymethylcytosine. *Nat Biotechnol* **29**, 68-72 (2011).
5. Suo, J. *et al.* VGLL4 promotes osteoblast differentiation by antagonizing TEADs-inhibited Runx2 transcription. *Sci Adv* **6**(2020).
6. Williams, K. *et al.* TET1 and hydroxymethylcytosine in transcription and DNA methylation fidelity. *Nature* **473**, 343-8 (2011).

REVIEWER COMMENTS

Reviewer #1 (Remarks to the Author):

Sufficient data is provided to show that Tets regulate occupancy of Runx2 including Tets interacting with Runx2. But this does mean Tets recruit Runx2 to their targets. Some ChIP-seq studies are needed, which I understand due to antibody issues cannot be done now. But authors have shown several targeted ChIP-qPCR which is sufficient. Authors should simply stay with the statement that Tets regulate occupancy of Runx2.

Just because any of the three Tets rescue phenotypes, doesn't mean that they occupy similar loci (it just means they compensate for each other, could be direct or indirect). This should be stated accurately.

Tet1/2/3 expression in TCKO cranial bone is very low while in TCKO femur is 20-30% (sup fig 2a-b) Please clarify? Non-specific deletion or presence of other cell types in femur?

Up regulated genes are likely gens that are kept repressed by Tets by virtue of interactions of Tets with chromatin repressive complexes like PRC2 and Sin3a etc... This can be included in discussion. This is more likely of an explanation than intragenic methylation.

Authors have made a strong effort to address my questions and manuscript maybe suitable for publication now.

Reviewer #2 (Remarks to the Author):

The revised manuscript has elevated the scientific quality of the study. Figure R2: it is difficult to see the Perilipin staining. Could the authors comment on Set redundancy in bone development is intriguing in terms of other studies showing that Tet3 seems to have no functional affect on human MSC and is lowly expressed by mouse MSC.

Reviewer #3 (Remarks to the Author):

The authors added the histological analysis of femurs and in vitro chondrocyte differentiation data in Supplementary Fig. 3. In their analysis, Col1a1 expression in bone collar was mildly reduced at E15.5, indicating that chondrocyte maturation was delayed in TCKO mice. However, the Col1a1 expression in bone collar in TCKO was comparable to WT at E16.5, although vascular invasion occurred in WT but not in TCKO and the trabecular bone formation was absent in TCKO. It indicates that bone collar, which is formed through intramembranous ossification, was normally formed and osteoblast differentiation in the bone collar was not impaired after chondrocyte maturation.

Safranin O staining at E13.5-16.5 also showed that chondrocyte maturation and vascular invasion into the cartilage were delayed in TCKO. However, Safranin O staining was absent in the bone marrow of TCKO at P0. Although the authors described that the bone marrow could be occupied by cartilage, it should be bone because there was no Safranin O staining. The TRAP staining in Supplementary figure 4 also shows the increase of bone in the bone marrow in TCKO. As the histological analysis from E16.5 to P0 was lacking, it is not clear when vascular invasion occurred and the bone formation in the bone marrow was accelerated in TCKO. In detailed analysis of skeletal development, including ISH of Col2a1, Col10a1, and Mmp13 for chondrocytes and Runx2 or Sp7, Col1a1, and Bglap for osteoblasts, is required for the understanding of the interesting phenotypes. Although the suture closure was delayed in TCKO mice, osteoblast differentiation may have been accelerated in limbs.

Although IHC of Edu was not clear, PCNA was apparently increased in the resting and proliferating

layers of TCKO and apoptosis was also increased in the cartilage. As the limbs in TCKO was shorter, it suggests that aberrantly accelerated cell cycle caused apoptosis of chondrocytes and resulted in dwarfism. Quantification of PCNA-positive cells, cleaved Cas3-positive cells, and the number of total chondrocytes will clarify this possibility. Chondrocyte apoptosis also impairs the chondrocyte differentiation in vitro. Moreover, the similar mechanism may have occurred in osteoblasts. Indeed, increased apoptosis also impairs bone formation in vivo and osteoblast differentiation in vitro. Thus, osteoblast proliferation and apoptosis in vivo need to be examined. As the authors focused on osteoblasts in calvaria, they should pay attention to the phenotypes of limbs, because they used Prx1 Cre, which deletes the target genes in mesenchymal cells, which can differentiate into chondrocytes and osteoblasts, in calvaria and limbs.

In Supplementary figure 4, TRAP-positive cells seemed to be increased in TCKO. If Runx2 function is impaired in TCKO, TRAP-positive cells will be decreased, because Rankl is one of the targets of Runx2. Rankl expression should be examined.

The length of clavicle in TCKO was slightly shorter than that in WT, whereas the clavicles were severely hypoplastic in Runx2+/- mice. It is also important to confirm whether Prx1-Cre is expressed in the clavicle region. TCKO mice showed severe dwarfism. Prx1 is expressed in calvaria and limbs but not in vertebrae. The shorted limbs were explained by Prx1 Cre, but the shortening of whole body is very hard to be explained. Are there any feeding problems? Finally, the phenotypes of limbs in TCKO were quite different from Runx2+/- mice. Although open fontanelle and sutures, which could be caused by many other factors, are the common phenotype, it is difficult to explain the phenotypes of TCKO by Runx2, as I mentioned in the first review. Runx2 may have been involved, in part, in the phenotypes but other genes would have majorly contributed to the phenotypes.

Reviewer #1 (Remarks to the Author):

Sufficient data is provided to show that Tets regulate occupancy of Runx2 including Tets interacting with Runx2. But this does mean Tets recruit Runx2 to their targets. Some ChIP-seq studies are needed, which I understand due to antibody issues cannot be done now. But authors have shown several targeted ChIP-qPCR which is sufficient. Authors should simply stay with the statement that Tets regulate occupancy of Runx2.

Just because any of the three Tets rescue phenotypes, doesn't mean that they occupy similar loci (it just means they compensate for each other, could be direct or indirect). This should be stated accurately.

Tet1/2/3 expression in TCKO cranial bone is very low while in TCKO femur is 20-30% (sup fig 2a-b) Please clarify? Non-specific deletion or presence of other cell types in femur?

Up regulated genes are likely genes that are kept repressed by Tets by virtue of interactions of Tets with chromatin repressive complexes like PRC2 and Sin3a etc... This can be included in discussion. This is more likely of an explanation than intragenic methylation.

Authors have made a strong effort to address my questions and manuscript maybe suitable for publication now.

Thank the reviewer for further suggestions to help us improve the manuscript. We agree with the reviewer that we should simply stay with the statement that TET proteins regulate occupancy of RUNX2. We modified the text accordingly in the revised manuscript. We also agree with the reviewer that 'any of the three Tets rescue phenotypes, doesn't mean that they occupy similar loci'. We have removed the TET1/2/3 form the same loci in the schematic diagram in the revised Fig.7k, and we just claimed that TET1/2/3 can compensate for each other, because they could bind to same loci or they could directly or indirectly affect the expression of same batch of genes.

Additionally, we have clarified the residual expression of *Tet1*, *Tet2*, and *Tet3* in the femurs of the TCKO mice in the revised M.S as "The residual expression of *Tet1*, *Tet2*, and *Tet3* in the femurs of the TCKO mice may be from the presence of other cell types in the femurs."

As the reviewer suggested, we added the discussion about the upregulated genes in TCKO cells in the revised M.S as "TET proteins have dual functions in the regulation of gene expression. TET mediated demethylation in the promoter region is correlated with active gene expression. Additionally, TET proteins involve the gene silencing by binding to certain regulatory factors and protein complexes. For example, TET proteins can interact with chromatin repressive complexes like PRC2 and SIN3A to repress gene expression^{1,2}. The upregulated genes in the TCKO cells are likely regulated by this manner."

Reviewer #2 (Remarks to the Author):

The revised manuscript has elevated the scientific quality of the study. Figure R2: it is difficult to see the Perilipin staining. Could the authors comment on Tet redundancy in bone development is intriguing in terms of other studies showing that Tet3 seems to have no functional affect on human MSC and is lowly expressed by mouse MSC.

Thank the reviewer for these suggestions. We increased the fluorescence brightness in Figure R2. As the figure showed, the accumulation of adipocytes in TCKO occurred at 9-week but not 2-week, suggesting that adipocytes accumulation may be a late stage effect of TET proteins in MSCs. As the reviewer suggested, we made a discussion about the function of TET1/2/3 in MSCs in the revised M.S as “Our current study demonstrates that loss of all three TET proteins in mesoderm mesenchymal stem cells results in severe bone defects and any single allele of *Tet* genes can rescue the defects of bone development. Another investigation demonstrated that *Tet1*, *Tet2* and *Tet3* were all expressed in both human and mouse BMMSCs, but only *Tet1* and *Tet2* decreased in the ovariectomized (OVX) mice derived BMMSCs³. In our study, *Tet1* and *Tet2* double conditional knockout mice showed subtle difference, compared with WT mice at P0, indicating that during the embryonic bone formation *Tet3* may provide compensated function in the absence of *Tet1* and *Tet2*. These observations indicate the functional compensation among three TET proteins during skeletal developmental process, similarly to the process of early gastrulation⁴.”

Reviewer #3 (Remarks to the Author):

The authors added the histological analysis of femurs and in vitro chondrocyte differentiation data in Supplementary Fig. 3. In their analysis, Col1a1 expression in bone collar was mildly reduced at E15.5, indicating that chondrocyte maturation was delayed in TCKO mice. However, the Col1a1 expression in bone collar in TCKO was comparable to WT at E16.5, although vascular invasion occurred in WT but not in TCKO and the trabecular bone formation was absent in TCKO. It indicates that bone collar, which is formed through intramembranous ossification, was normally formed and osteoblast differentiation in the bone collar was not impaired after chondrocyte maturation.

Thank the reviewer for this careful analysis. We repeated some histological analysis and confirmed that the chondrocyte maturation was delayed in TCKO mice as determined by Safranin O staining at E13.5-17.5 (Supplementary Figure 3a). The reviewer noticed that Col1a1 expression in bone collar was mildly reduced at E15.5, indicating that chondrocyte maturation was delayed in TCKO mice. However, the Col1a1 expression in bone collar in TCKO was comparable to WT at E16.5, although vascular invasion occurred in WT but not in TCKO and the trabecular bone formation was absent in TCKO. In this experiment, the ISH signal of Col1a1 expression is strong and the marginal difference of their expression level between WT and TCKO mice could be hard to be distinguished. Bglap is another marker for the later stage of osteoblast differentiation. We did immunofluorescence of Bglap for E16.5 mice and E17.5 mice found that the expression of Bglap was decreased in the bone collar of

TCKO mice at both E16.5 and E17.5 (Supplementary Figure 3e-3f), suggesting that the maturation of osteoblasts in the bone collar was impaired or delayed in TCKO mice.

Safranin O staining at E13.5-16.5 also showed that chondrocyte maturation and vascular invasion into the cartilage were delayed in TCKO. However, Safranin O staining was absent in the bone marrow of TCKO at P0. Although the authors described that the bone marrow could be occupied by cartilage, it should be bone because there was no Safranin O staining. The TRAP staining in Supplementary figure 4 also shows the increase of bone in the bone marrow in TCKO. As the histological analysis from E16.5 to P0 was lacking, it is not clear when vascular invasion occurred and the bone formation in the bone marrow was accelerated in TCKO. In detailed analysis of skeletal development, including ISH of Col2a1, Col10a1, and Mmp13 for chondrocytes and Runx2 or Sp7, Col1a1, and Bglap for osteoblasts, is required for the understanding of the interesting phenotypes. Although the suture closure was delayed in TCKO mice, osteoblast differentiation may have been accelerated in limbs.

Thank the review for these suggestions. The reviewer suggested that the bone marrow was occupied by bone, but not cartilage as there was no Safranin O staining. Our recent *Cell Stem Cell* paper demonstrated that postnatal osteoblasts arose primarily from chondrocytes before adolescence and from *Lepr*⁺ bone marrow stromal cells (BMSCs) after adolescence (*Cell Stem Cell*, 2021, 28: 2122-2136 Tracing the skeletal progenitor transition during postnatal bone formation). The absence of Safranin O staining in the bone marrow of TCKO at P0 could be a situation during the transition of chondrocytes to osteoblasts. We then performed Safranin O staining at E17.5 and found that the Safranin O staining is absent in the bone marrow of WT mice but is present in the TCKO mice (Supplementary Fig. 3a). These data suggested the delay of the transition from chondrocytes to osteoblasts (endochondral ossification) in the bone marrow of TCKO mice. Our data did not support that 'the bone formation in the bone marrow was accelerated in TCKO'.

Following the reviewer's suggestion, we also performed the analysis of Col10a1 and Mmp13 for chondrocytes and Sp7 and Bglap for osteoblasts at E16.5 and/or E17.5 (Supplementary Fig. 3b-3g). The expression of Bglap and Osx was decreased in the femur of TCKO mice (Supplementary Fig. 3e-3g). However, more Col10a1 and Mmp13 expression was seen in the bone marrow of TCKO mice compared with WT mice at E17.5 (Supplementary Fig. 3b-3d), suggesting the delay of the transition from chondrocytes to osteoblasts (endochondral ossification) in the bone marrow of TCKO mice. Again, these data supported that loss of TET proteins delayed osteoblast maturation during the endochondral bone formation but did not support that 'the bone formation in the bone marrow was accelerated in TCKO'.

Supplemental Figure 3. Loss of TET enzymes disrupt the endochondral bone formation.

(a) Safranin O staining of limbs isolated from WT and TCKO mice at different developmental stages. Scale bar = 500µm. (b-g) Immunofluorescence of Col10a1 at E16.5 (b) and E17.5 (c), Mmp13 at E17.5 (d), Bglap at E16.5 (e) and E17.5 (f), and Osx (g) in the femurs of WT and TCKO mice. Scale bar = 200µm. (h) Immunofluorescence of cleaved-caspase 3 in the femurs of WT and TCKO mice at P0. Scale bar: 500µm for the top images and 40µm for the bottom images. (i) Immunofluorescence of proliferation marker PCNA in the femurs of WT and TCKO mice at P0. Scale bar = 500µm. (j) Quantification of PCNA-positive cells, cleaved Cas3-positive cells, and the number of total chondrocytes in (h-i).

Although IHC of Edu was not clear, PCNA was apparently increased in the resting and proliferating layers of TCKO and apoptosis was also increased in the cartilage. As the limbs in TCKO was shorter, it suggests that aberrantly accelerated cell cycle caused apoptosis of chondrocytes and resulted in dwarfism. Quantification of PCNA-positive cells, cleaved Cas3-positive cells, and the number of total chondrocytes will clarify this possibility. Chondrocyte apoptosis also impairs the chondrocyte differentiation in vitro. Moreover, the similar mechanism may have occurred in osteoblasts. Indeed, increased apoptosis also impairs bone formation in vivo and osteoblast differentiation in vitro. Thus, osteoblast proliferation and apoptosis in vivo need to be examined. As the authors focused on osteoblasts in calvaria, they should pay attention to the phenotypes of limbs, because they used Prx1 Cre, which deletes the target genes in mesenchymal

cells, which can differentiate into chondrocytes and osteoblasts, in calvaria and limbs.

Thank the review for these suggestions. As the reviewer suggested, we quantified the PCNA-positive cells, cleaved Cas3-positive cells, and the number of total chondrocytes (Supplementary Fig. 3h-3j). We found the increased PCNA-positive cells and increased cleaved Cas3-positive cells. The difference could be due to that TCKO mice stayed a delayed stage compared to the WT mice. We also found less chondrocytes in the TCKO mice, which could contribute to the dwarfism and could be due to the dysregulated cell cycle. We also examined the PCNA-positive cells, cleaved Cas3-positive cells in the cranial bones and found no significant changes between WT and TCKO mice (Figure R3b-3c).

Figure R3. Loss of TET enzymes disrupt the intramembranous bone formation. (a) Representative image by fluorescence microscopy from the clavicle bone shown Prx1 positive cells using *Prx1-Cre, Ai9/+* mice. Scale bar = 200 μm. (b) Immunofluorescence of proliferation marker PCNA in the femurs of WT and TCKO mice at P0. Scale bar = 50 μm. (c) Immunofluorescence of cleaved-caspase 3 in the femurs of WT and TCKO mice at P0. Scale bar: 50 μm.

In Supplementary figure 4, TRAP-positive cells seemed to be increased in TCKO. If Runx2 function is impaired in TCKO, TRAP-positive cells will be decreased, because Rankl is one of the targets of Runx2. Rankl expression should be examined.

Thank the review for these suggestions. As the reviewer suggested, we examined the expression of *Rankl* in the long bones of WT and TCKO mice and found that *Rankl* were not significantly affected by the loss TET enzymes (Supplementary Fig. 4b). The difference of TRAP staining pattern, if any, could be due to the delayed endochondral bone formation in TCKO mice.

b

Supplemental Figure 4. Loss of TET enzymes have no significant effects on bone remodeling after birth. (b) RT-qPCR analysis of *Rankl* expression in the long bones of WT and TCKO mice. * $P < 0.05$, ** $P < 0.01$. Two-tailed Student's t-test. Data are presented as mean \pm s.d., $n=4$.

The length of clavicle in TCKO was slightly shorter than that in WT, whereas the clavicles were severely hypoplastic in *Runx2*^{+/-} mice. It is also important to confirm whether *Prx1*-Cre is expressed in the clavicle region. TCKO mice showed severe dwarfism. *Prx1* is expressed in calvaria and limbs but not in vertebrae. The shorted limbs were explained by *Prx1* Cre, but the shortening of whole body is very hard to be explained. Are there any feeding problems? Finally, the phenotypes of limbs in TCKO were quite different from *Runx2*^{+/-} mice. Although open fontanelle and sutures, which could be caused by many other factors, are the common phenotype, it is difficult to explain the phenotypes of TCKO by *Runx2*, as I mentioned in the first review. *Runx2* may have been involved, in part, in the phenotypes but other genes would have majorly contributed to the phenotypes.

As the reviewer suggested, we did cell fate tracing experiment by examining the existence of *Ai9* in the clavicle of *Prx1*-Cre; *Ai9*⁺ mice. We can see the existence of *Ai9* positive cells in the clavicle bone (Figure R3a). Our study demonstrated that TET could directly interact with RUNX2 and regulate RUNX2 activity. The shorter clavicle bone of TCKO mice could be due to the decreased RUNX2 activity. The length of clavicle in TCKO was slightly shorter than that in WT, whereas the clavicles were severely hypoplastic in *Runx2*^{+/-} mice. As we know, the clavicle bone is very sensitive to the defects of RUNX2 activity. However, it is a common situation that the deletion of RUNX2 regulators did not display severely hypoplastic clavicles. For example, *Prx1*-cre *Med23*^{ff} mice. It is possible that during the early development, the fine-tune regulation of RUNX2 activity by these regulators, including TETs, happens later than the moment that RUNX2 functions to regulate clavicle bone formation.

The reviewer suggested that the shortening of whole body is very hard to be explained as *Prx1* is expressed in calvaria and limbs but not in vertebrae. Actually, the deletion of certain genes by *Prx1*-Cre could lead to the shorter mice. For example, *Prx1*-Cre *NF1*^{ff} mice (*Hum Mol Genet*, 2007, 16, 874-886. Multiple roles for neurofibromin in

skeletal development and growth), *Prx1-cre Med23^{ff}* mice (*Nature communications*, 2016, 7, 11149. Mediator MED23 cooperates with RUNX2 to drive osteoblast differentiation and bone development), *Prx1-cre Stat3^{ff}* mice (*Nature communications*, 2021, 12, 6891. STAT3 is critical for skeletal development and bone homeostasis by regulating osteogenesis), etc⁵⁻⁷ (Figure R4a-4c). These could be an early developmental issue or the coordination to the defects in the calvaria and limbs.

The TCKO mice showed severe dwarfism. The shortening of whole body of 3-week old mice could be due to the feeding difficulty, as the development stage of TCKO mice were delayed and the mice were shorter and weaker at birth compared to the WT mice. We agree with the reviewer that other genes would contribute to this phenotype, as the TCKO mice showed strong defects in the bone. As the Reviewer 1 suggested, there are bench of upregulated genes in TCKO cells and TET mediated demethylation in the promoter region could be also correlated with active gene expression. The up-regulated genes in the TCKO cells may also contribute to the complicated phenotype in TCKO mice. We have added some discussion into the revised manuscript.

In summary, the reviewer's concerns focus on the possibility that other mechanisms besides RUNX2 contributes to the bone developmental phenotype we observed in our *Tets* TCKO mice. Our study demonstrated that TET proteins could interact with RUNX2 and regulate RUNX2 occupancy on the promoters of RUNX2 downstream genes. *Tets* TCKO mice displayed all of the RUNX2 related phenotypes including open fontanelle and sutures, shorter clavicles and delayed maturation of bone in the limbs. As RUNX2 is the key transcription factor for bone development, we believe that the regulation of RUNX2 by TETs is important for bone phenotype of TCKO mice. The reviewer's comments 'the phenotypes of limbs in TCKO were quite different from *Runx2*^{+/-} mice' might because he/or she incorrectly interpreted the phenotype of our TCKO mice as 'the bone formation in the bone marrow was accelerated in TCKO'. However, our data suggested a delay of the bone formation in the bone marrow of TCKO mice as we mentioned above. The reviewer also concerns that the short body length and short limb length phenotype in our TCKO mice. We would like to pointed out that although the body length and limb length of *Runx2*^{+/-} mice are not significant different with WT mice, our recent study demonstrated that *Acan CreER-Runx2^{ff}* mice displayed short body length and limb length (*Cell Stem Cell*, 2021, 28: 2122-2136 Tracing the skeletal progenitor transition during postnatal bone formation) (Figure R4d), indicating that the dysregulation of Runx2 could lead to shorter body length and limb length. The reviewer also concerns that the shortening of whole body is very hard to be explained as Prx1 is expressed in calvaria and limbs but not in vertebrae. Actually, the deletion of certain genes by *Prx1-Cre* could lead to the shorter mice as we listed in (Figure R4a-c).

Figure R4 (a-d) Representative view of (a) *Prx1-Cre NF1^{fl/fl}* mice (*Hum Mol Genet*, 2007, 16, 874-886. Multiple roles for neurofibromin in skeletal development and growth), (b) *Prx1-cre Med23^{fl/fl}* mice (*Nature communications*, 2016, 7, 11149. Mediator MED23 cooperates with RUNX2 to drive osteoblast differentiation and bone development), (c) *Prx1-cre Stat3^{fl/fl}* mice (*Nature communications*, 2021, 12, 6891. STAT3 is critical for skeletal development and bone homeostasis by regulating osteogenesis) and (d) *Acan-creER; Runx2^{fl/fl}* mice (*Cell Stem Cell*, 2021, 28: 2122-2136 Tracing the skeletal progenitor transition during postnatal bone formation).

Reference

1. Yang, J., Bashkenova, N., Zang, R., Huang, X. & Wang, J. The roles of TET family proteins in development and stem cells. *Development* **147**(2020).
2. Williams, K. *et al.* TET1 and hydroxymethylcytosine in transcription and DNA methylation fidelity. *Nature* **473**, 343-8 (2011).
3. Yang, R. *et al.* Tet1 and Tet2 maintain mesenchymal stem cell homeostasis via demethylation of the P2rx7 promoter. *Nat Commun* **9**, 2143 (2018).
4. Dai, H.Q. *et al.* TET-mediated DNA demethylation controls gastrulation by regulating Lefty-Nodal signalling. *Nature* **538**, 528-532 (2016).
5. Zhou, S. *et al.* STAT3 is critical for skeletal development and bone homeostasis by regulating osteogenesis. *Nat Commun* **12**, 6891 (2021).
6. Liu, Z. *et al.* Mediator MED23 cooperates with RUNX2 to drive osteoblast differentiation and bone development. *Nat Commun* **7**, 11149 (2016).
7. Kolanczyk, M. *et al.* Multiple roles for neurofibromin in skeletal development and growth. *Hum Mol Genet* **16**, 874-86 (2007).

REVIEWER COMMENTS

Reviewer #3 (Remarks to the Author):

Although the authors responded my concerns and added the data, the data and the comprehension of the data have many problems.

1. Osx-positive cells in TCKO mice at E17.5 are few. It indicates that vascular invasion is absent in this section and the diaphysis is occupied by the terminally differentiated chondrocytes. Indeed, osteoblast differentiation in the bone collar should be delayed as shown by the Bglap expression due to the delayed chondrocyte maturation in TCKO mice. The process of the endochondral ossification (Fig. S3a) shows the delay in chondrocyte maturation but not the delay in the transition of chondrocytes to osteoblasts. It is not an appropriate age to examine osteoblast differentiation without the effect of chondrocyte maturation.

2. Safranin O staining at P0 shows a lot of fast green-stained matrix in the bone marrow and bone collar (Supplementary figure 3). These are bone matrix. In The TRAP staining in Sup Fig. 4, bone matrix is also apparent in TCKO mice. This is not a situation during the transition of chondrocytes to osteoblasts. Abundant bone is formed in TCKO mice at P0. The authors can confirm it by ISH using osteoblast marker genes at P0.

3. The increase of PCNA-positive cells and cleaved Cas3-positive cells is not due to the delayed chondrocyte maturation. The accelerated cell cycle will not only increase apoptosis but also delay the chondrocyte maturation. This is quite different from the delay of chondrocyte maturation in Runx2 ko mice, in which there are no increase of proliferative marker and apoptosis. Further, the delay in chondrocyte maturation is not apparent in Runx2^{+/-} mice. Thus, the phenotypes in chondrocytes are not related to Runx2.

4. Is the Figure R3a really clavicle? The shape is strange. The medial part of clavicle is formed through endochondral ossification and the lateral part of clavicle is formed through intramembranous ossification. The shortening of the clavicle is more likely to be explained by the increased apoptosis due to the accelerated cell cycle in chondrocytes. The legend of Fig. R3b and 3c indicates femur and the response indicates cranial bones. Which area are magnified? It cannot be concluded that the proliferation and apoptosis are normal in TCKO osteoblasts from these pictures. It is very important, because the increased apoptosis impairs osteoblast differentiation in vitro and delays the calvarial development.

5. The dwarfism will be mainly due to the feeding problem. The severity of the shortness of the limbs will be related to the severity of the dwarfism. It also indicates that the length of the limbs was severely affected in TCKO mice. It is the most prominent phenotype of TCKO mice and not related to Runx2. The authors claimed that Tets TCKO mice displayed all of the RUNX2 related phenotypes including open fontanelle and sutures, shorter clavicles and delayed maturation of bone in the limbs. The open fontanelle and sutures in TCK mice are much milder than those in Runx2^{+/-} mice, and could be explained by other genes or aberrantly enhanced cell cycle in osteoblasts by Tet. The shorter clavicles are likely due to the increased apoptosis as mentioned above. The mechanism in the delayed maturation of bone in the limbs is completely different from that in Runx2 ko mice.

6. I did not think that bone formation in the bone marrow is accelerated, because it is unlikely. However, the previous data of Safranin O at P0 in TCKO mice showed many bone matrix than wild-type mice. The new Safranin O and fast green staining also showed abundant bone matrix. Therefore, Bone formation is likely to be not apparently impaired in TCKO mice in limbs. The transition of chondrocytes to osteoblasts is not impaired from the histological section at P0.

7. The authors showed that Acan CreER-Runx2^{fl/fl} mice display shortened body length and limbs in the previous paper (Cell Stem Cell). As Acan Cre deletes Runx2 in immature chondrocytes, it impairs the chondrocyte maturation and leads to shortened body length and limbs. It was shown in many papers and the mechanism is completely different from that in TCKO mice.

Although the authors responded my concerns and added the data, the data and the comprehension of the data have many problems.

1. *Osx*-positive cells in TCKO mice at E17.5 are few. It indicates that vascular invasion is absent in this section and the diaphysis is occupied by the terminally differentiated chondrocytes. Indeed, osteoblast differentiation in the bone collar should be delayed as shown by the *Bglap* expression due to the delayed chondrocyte maturation in TCKO mice. The process of the endochondral ossification (Fig. S3a) shows the delay in chondrocyte maturation but not the delay in the transition of chondrocytes to osteoblasts. It is not an appropriate age to examine osteoblast differentiation without the effect of chondrocyte maturation.

The reviewer observed the *Osx*-positive cells are few in TCKO mice at E17.5. The observation of less *Osx*-positive cells in TCKO mice is consistent with the decreased *RUNX2* activity and decreased osteoblast differentiation as *RUNX2* could directly regulate the expression of *Osx*. We also observed that *Osx* was less expressed in the cortical bone in TCKO mice at E17.5 (Fig. S3f-3g), indicating that the bone formation ability was attenuated in the TCKO mice.

Following the reviewer's suggestion during the last round of revision, we examined the expression of *Col10a1* and *Mmp13* for chondrocytes and *Sp7* (*Osx*) and *Bglap* for osteoblasts at E16.5 and/or E17.5 (Supplementary Fig. 3b-3g). The expression of *Bglap* and *Osx* was decreased in the femur of TCKO mice (Supplementary Fig. 3e-3g). However, more *Col10a1* and *Mmp13* expression was seen in the bone marrow of TCKO mice compared with WT mice at E17.5 (Supplementary Fig. 3b-3d). These data supported that loss of TET proteins delayed osteoblast maturation during the endochondral bone formation but did not support the reviewer's last conclusion that 'the bone formation in the bone marrow was accelerated in TCKO'.

2. Safranin O staining at P0 shows a lot of fast green-stained matrix in the bone marrow and bone collar (Supplementary figure 3). These are bone matrix. In The TRAP staining in Sup Fig. 4, bone matrix is also apparent in TCKO mice. This is not a situation during the transition of chondrocytes to osteoblasts. Abundant bone is formed in TCKO mice at P0. The authors can confirm it by ISH using osteoblast marker genes at P0.

A mixture of Safranin O and fast green staining in the bone marrow was shown at P0 (Fig. S3a). The reviewer named them as 'bone matrix'. We think this is immature 'bone matrix' as we discussed above. A serial of histological staining of E13.5, E15.5, E16.5, E17.5 and P0 indicated that the delay of bone formation (Supplemental Figure 3). The reviewer suggested the ISH using osteoblast marker genes at P0. Following the reviewer's last round of suggestions, we examined the expression of *Col10a1* and *Mmp13* for chondrocytes and *Sp7* (*Osx*) and *Bglap* for osteoblasts at E16.5 and/or E17.5 (Supplementary Fig. 3b-3g). The expression of *Bglap* and *Osx* was decreased in the femur of TCKO mice (Supplementary Fig. 3e-3g). However, more *Col10a1* and *Mmp13* expression was seen in the bone marrow of TCKO mice compared with WT mice at E17.5 (Supplementary Fig. 3b-3d). The data is consistent with the delay of

bone formation in TCKO mice.

Supplemental Figure 3. Loss of TET enzymes disrupt the endochondral bone formation.

(a) Safranin O staining of limbs isolated from WT and TCKO mice at different developmental stages. Scale bar = 500µm. (b-g) Immunofluorescence of Col10a1 at E16.5 (b) and E17.5 (c), Mmp13 at E17.5 (d), Bglap at E16.5 (e) and E17.5 (f), and Osx (g) in the femurs of WT and TCKO mice. Scale bar = 200µm. (h) Immunofluorescence of cleaved-caspase 3 in the femurs of WT and TCKO mice at P0. Scale bar: 500µm for the top images and 40µm for the bottom images. (i) Immunofluorescence of proliferation marker PCNA in the femurs of WT and TCKO mice at P0. Scale bar = 500µm. (j) Quantification of PCNA-positive cells, cleaved Cas3-positive cells, and the number of total chondrocytes in (h-i).

3. The increase of PCNA-positive cells and cleaved Cas3-positive cells is not due to the delayed chondrocyte maturation. The accelerated cell cycle will not only increase apoptosis but also delay the chondrocyte maturation. This is quite different from the delay of chondrocyte maturation in Runx2 ko mice, in which there are no increase of proliferative marker and apoptosis. Further, the delay in chondrocyte maturation is not apparent in Runx2^{+/-} mice. Thus, the phenotypes in chondrocytes are not related to Runx2.

We agree with the reviewer that other genes would contribute to this phenotype. As the Reviewer 1 suggested, there are bench of upregulated genes in TCKO cells and TET mediated demethylation in the promoter region could be also correlated with active gene expression. The up-regulated genes in the TCKO cells may also contribute to the complicated phenotype in TCKO mice. We have added some discussion into the last version of revised manuscript L252-258. What we mean 'chondrocyte maturation' is 'the transition of chondrocyte to osteoblast' or 'endochondral ossification' (*Cell Stem Cell*, 2021, 28: 2122-2136 Tracing the skeletal progenitor transition during postnatal bone formation). We made a correction in the revised manuscript to make it clear. Additionally, *Prx1-cre-Runx2^{fl/fl}* mice also showed the impaired endochondral ossification (Fig. R5) (*Development*, 2015, 143(2):211-8. Genetic analysis of Runx2 function during intramembranous ossification).

Figure R5. The skeleton of wild-type WT and *Runx2^{prx1-/-}* mice at E18.5 (DOI: 10.1242/dev.128793).

4. Is the Figure R3a really clavicle? The shape is strange. The medial part of clavicle is formed through endochondral ossification and the lateral part of clavicle is formed through intramembranous ossification. The shortening of the clavicle is more likely to be explained by the increased apoptosis due to the accelerated cell cycle in chondrocytes. The legend of Fig. R3b and 3c indicates femur and the response indicates cranial bones. Which area are magnified? It cannot be concluded that the proliferation and apoptosis are normal in TCKO osteoblasts from these pictures. It is very important, because the increased apoptosis impairs osteoblast differentiation in vitro and delays the calvarial development.

This is clavicle at postnatal day 10. The *Prx1-cre* can label both endochondral ossification and intramembranous ossification. Fig. R3b and 3c is the cranial bone, we have corrected it in the revised version. Additionally, the apoptosis of osteoblasts was not affected in TCKO mice in vitro (Fig. R3d-3e), demonstrating that the impaired osteoblast differentiation in vitro was not caused by apoptosis.

Figure R3. Loss of TET enzymes disrupt the intramembranous bone formation. (a) Representative image by fluorescence microscopy from the clavicle bone shown Prx1 positive cells using *Prx1-Cre, Ai9/+* mice. Scale bar = 200 μm . (b) Immunofluorescence of proliferation marker PCNA in the cranial bone of WT and TCKO mice at P0. Scale bar = 50 μm . (c) Immunofluorescence of cleaved-caspase 3 in the cranial bone of WT and TCKO mice at P0. Scale bar: 50 μm . (d-e) Apoptosis analysis by FACS of Annexin V signals of osteoblasts isolated from the cranial bone of WT and TCKO mice, n=2.

5. The dwarfism will be mainly due to the feeding problem. The severity of the shortness of the limbs will be related to the severity of the dwarfism. It also indicates that the length of the limbs was severely affected in TCKO mice. It is the most prominent phenotype of TCKO mice and not related to Runx2. The authors claimed that Tets TCKO mice displayed all of the RUNX2 related phenotypes including open fontanelle and sutures, shorter clavicles and delayed maturation of bone in the limbs. The open fontanelle and sutures in TCK mice are much milder than those in Runx2 $^{+/-}$ mice, and could be explained by other genes or aberrantly enhanced cell cycle in osteoblasts by Tet. The shorter clavicles are likely due to the increased apoptosis as mentioned above. The mechanism in the delayed maturation of bone in the limbs is completely different from that in Runx2 ko mice.

The reviewer thinks the shortness of the limbs is the most prominent phenotype of TCKO mice and it is not related to Runx2. We disagree with this. We would like to point out that the dysregulation of Runx2 could induce the shortness of the limbs although Runx2 $^{+/-}$ mice barely display shortened body length and limbs. Actually, *Acan^{CreER}-Runx2^{fl/fl}* showed shortened body length and limbs in our previous paper (Cell Stem Cell). More importantly, *Prx1-cre-Runx2^{fl/fl}* mice also showed dramatic shortening of whole body (*Development*, 2015, 143(2):211-8. Genetic analysis of Runx2 function during intramembranous ossification).

The cell cycle of cranial bone was not affected in TCKO mice (Fig. R3b-3e). We performed the motif analysis over hyper-CG-DMR in TCKO and we identified the top five enriched known motifs, including Runt, CTF, bZIP, TEA/TEAD, RHD, and bHLH families (Fig. 5c). And then we found that RUNX2 can interact with all three TET proteins (Fig. 6a-6b), and loss of TETs affects the binding of RUNX2 to its targeted genes (Fig. 7e-7g). Although the severity of phenotype was not completely equal, TCKO mice displayed all of the RUNX2 related phenotypes including open fontanelle and sutures, shorter clavicles and delayed maturation of bone in the limbs. As RUNX2 is the key transcription factor for bone development, we believe that the regulation of RUNX2 by TETs is important for bone phenotype of TCKO mice.

The reviewer concerned that 'The open fontanelle and sutures in TCKO mice are much milder than those in *Runx2*^{+/-} mice, and could be explained by other genes or aberrantly enhanced cell cycle in osteoblasts by Tet'. Similar to the length of clavicle, the open fontanelle and sutures in TCKO were milder than that in *Runx2*^{+/-} mice. As we know, the open fontanelle and sutures and the length of clavicle bone are very sensitive to the defects of RUNX2 activity. However, it is a common situation that the deletion of RUNX2 regulators did not display severely phenotype as *Runx2*^{+/-} mice. For example, *Prx1-cre Med23*^{fl/fl} mice. It is possible that during the early development, the fine-tune regulation of RUNX2 activity by these regulators, including TETs, happens later than the moment that RUNX2 functions to regulate early bone formation.

6. I did not think that bone formation in the bone marrow is accelerated, because it is unlikely. However, the previous data of Safranin O at P0 in TCKO mice showed many bone matrix than wild-type mice. The new Safranin O and fast green staining also showed abundant bone matrix. Therefore, Bone formation is likely to be not apparently impaired in TCKO mice in limbs. The transition of chondrocytes to osteoblasts is not impaired from the histological section at P0.

During the last round of the review, the reviewer suggested that 'bone formation in the bone marrow is accelerated' as the 'bone matrix' was abundant in P0 of TCKO mice. The 'bone matrix' in P0 of TCKO is a mixture of Safranin O and fast green staining (Fig. S3a), which we suggested to be an immature bone. Our histological staining of E13.5, E15.5, E16.5, E17.5 and P0 indicated that the delay of bone formation (Supplemental Figure 3, see above). During the last round of revision, we also examined the expression of Col10a1 and Mmp13 for chondrocytes and Sp7 (Osx) and Bglap for osteoblasts at E16.5 and/or E17.5 (Supplementary Fig. 3b-3g). The expression of Bglap and Osx, which are regulated by RUNX2, were decreased in the femur of TCKO mice (Supplementary Fig. 3e-3g). However, more Col10a1 and Mmp13 expression was seen in the bone marrow of TCKO mice compared with WT mice at E17.5 (Supplementary Fig. 3b-3d). These data supported that loss of TET proteins delayed osteoblast maturation during the endochondral bone formation but did not support that 'the bone formation in the bone marrow was accelerated in TCKO'. These data again did not support the reviewer's new 'conclusion' that 'bone formation is likely to be not apparently impaired in TCKO mice in limbs'.

7. The authors showed that *Acan* CreER-*Runx2*^{fl/fl} mice display shortened body length and limbs in the previous paper (Cell Stem Cell). As *Acan* Cre deletes *Runx2* in immature chondrocytes, it impairs the chondrocyte maturation and leads to shortened body length and limbs. It was shown in many papers and the mechanism is completely different from that in TCKO mice.

During the last round of the review, the reviewer pointed out that the phenotype of TCKO mice are different from *Runx2*^{+/-} mice as the *Prx1-cre* TCKO showed dwarfism. The reviewer suggested that the shortening of whole body is very hard to be explained as *Prx1* is expressed in calvaria and limbs but not in vertebrae. We explained this point and showed that the deletion of certain genes by *Prx1-Cre* could lead to the shorter mice. For example, *Prx1-Cre NF1*^{ff} mice (*Hum Mol Genet*, 2007, 16, 874-886. Multiple roles for neurofibromin in skeletal development and growth), *Prx1-cre Med23*^{ff} mice (*Nature communications*, 2016, 7, 11149. Mediator MED23 cooperates with RUNX2 to drive osteoblast differentiation and bone development), *Prx1-cre Stat3*^{ff} mice (*Nature communications*, 2021, 12, 6891. STAT3 is critical for skeletal development and bone homeostasis by regulating osteogenesis), etc⁵⁻⁷ (Figure R4a-4c). We also pointed out that although *Runx2*^{+/-} mice did not display shortened body length and limbs, *Acan*^{CreER}-*Runx2*^{fl/fl} showed shortened body length and limbs in our previous paper (Cell Stem Cell). This time, the reviewer's new comments is that 'As *Acan* Cre deletes *Runx2* in immature chondrocytes.... the mechanism is completely different from that in TCKO mice'. We would like to point out that *Prx1-cre* can also label the immature chondrocytes (or the osteoblast progenitor cells, or the skeletal stem cells). Additionally, *Prx1-cre-Runx2*^{fl/fl} mice also showed the shortening of whole body, the open fontanelle and sutures, short clavicle, etc (Fig. R5) (*Development*, 2015, 143(2):211-8. Genetic analysis of *Runx2* function during intramembranous ossification). The phenotype of our TCKO mice could be explained by the dysregulation of RUNX2.

REVIEWER COMMENTS

Reviewer #4 (Remarks to the Author):

The author addressed the reviewer's concerns, but the author didn't have adequately addressed them. For example, these following concerns that are not well answered:

1, Comments, No.5, "The dwarfism will be mainly due to the feeding problem. The severity of the shortness of the limbs will be related to the severity of the dwarfism. It is the most prominent phenotype of TCKO mice and not related to Runx2. The shorter clavicles are likely due to the increased apoptosis as mentioned." The relevant research mechanism should be explored.

2, Comments, No.6, "I did not think that bone formation in the bone marrow is accelerated, because it is unlikely. However, the previous data of Safranin O at P0 in TCKO mice showed many bone matrix than wild-type mice. The new Safranin O and fast green staining also showed abundant bone matrix. Therefore, Bone formation is likely to be not apparently impaired in TCKO mice in limbs. The transition of chondrocytes to osteoblasts is not impaired from the histological section at P0". This comment was concerned with bone formation and transition of chondrocytes to osteoblasts at P0. While the author's new data focused on E16.5 and/or E17.5, not P0. Thus, the author's reply to this comment was inappropriate and insufficient.

Reviewer #5 (Remarks to the Author):

In this paper, the authors suggest that TET enzymes function to regulate RUNX2 activity and play a role in a bone formation, especially in the differentiation of osteoblast both in calvarial bones (via intramembranous ossification) and femurs (via endochondral ossification). These results will be helpful to unravel molecular events occurring by TET enzyme in the bone development. The data are interesting and add to the literature. There are some shortcomings that could be addressed:

1. The authors used Prx1-Cre:Tet conditional knockout mice to show the impairment of bone formation and furthermore, they also showed that Tet interacts with RUNX2 to regulate osteogenesis. In limb, beside osteoprogenitors, chondroprogenitors are also most likely expressed Prx1. As well, RUNX2 is also expressed in chondrocyte and direct target genes of RUNX2 in chondrocytes are *Ihh*, *Col10a1*, and *Mmp13* in the limb. Moreover, it has been know that in the case of RUNX2 KO, chondrocyte-specific KO of RUNX2 induced severe dwarfism. However, bone-specific KO did not induce severe dwarfism compared to chondrocyte-specific KO of RUNX2. It is rather prevent the transition of cartilage to bone formation. Therefore, the process of the endochondral ossification (Supplementary Figure 3) more likely due to the delay in chondrocyte maturation (Chondrocyte hypertrophy) but not the delay in the transition of chondrocytes to osteoblasts (or osteoblast maturation).

2. In addition, since Prx1 interacts with RUNX2 as authors showed, there is high possibility that hypertropic chondrocytes in the limb might be affected (Authors showed that direct target genes of RUNX2 in chondrocytes, *COL10a1* and *MMP13* had affected as well in the paper...pointing that it is rather chondrocyte maturation affected by Tet KO).

Since Prx1 and RUNX are expressed in both osteoblast and chondrocyte, the authors should interpret carefully the role of Prx1-Cre:Tet and RUNX2 during the process of endochondral ossification.

REVIEWER COMMENTS

Reviewer #4 (Remarks to the Author):

The author addressed the reviewer's concerns, but the author didn't have adequately addressed them. For example, these following concerns that are not well answered:

1, Comments, No.5, "The dwarfism will be mainly due to the feeding problem. The severity of the shortness of the limbs will be related to the severity of the dwarfism. It is the most prominent phenotype of TCKO mice and not related to Runx2. The shorter clavicles are likely due to the increased apoptosis as mentioned." The relevant research mechanism should be explored.

Thanks a lot for the reviewer's comments and suggestions. TCKO mice exhibited short limb with some defects in feeding after birth, which may lead to a more severe and systemically dwarfism in its adult life. We did not agree with the previous reviewer 3 that the phenotype is not related to the Runx2 as *Prx1-Cre*, *Runx2^{fl/fl}* mice displayed the shortness of the limbs and the dwarfism¹. For the relevant research mechanism, the reviewer 5 suggested that TCKO mice have the defects in endochondral ossification as RUNX2 target genes, including *COI10a1* and *MMP13* were also affected in TCKO mice (Supplementary Fig. 3b-3d). We agreed with this as *Prx1-cre* could label both chondroprogenitor cells and osteoprogenitor cells. Actually, our high-throughput sequencing data also indicated that both *COI10a1* and *MMP13* were occupied by RUNX2 (Supplementary Fig. 10g-10f). Loss of TET proteins decreased the chromatin accessibility of *COI10a1* and *MMP13* with increased 5mC level and decreased 5hmC level (Supplementary Fig. 10g-10f). These data indicated that the interaction of TET proteins with RUNX2 could also regulate the genes related to the chondrocytes. However, the detailed mechanism how TET proteins regulated the gene expression of *COI10a1* and *MMP13* is worthy to be investigated in the further studies. We added this discussion in the revised manuscript line 390-403.

Supplementary Fig 10. (g-f) Screenshot of *Col10a1* and *Mmp13* in CG methylation level from WGBS, 5hmC-Seal, ATAC-seq in WT and TCKO primary osteoblasts.

The clavicle is formed by both intramembranous and endochondral ossification. The defects of clavicle are mostly caused by the dysfunction of RUNX2 and its coregulators,

including Med23², Vgl14³, etc. As TET proteins can interact with RUNX2 and regulate the expression of RUNX2 targeted genes, loss of TET proteins could impair the function of RUNX2 in both endochondral and intramembranous bone formation. The shorter clavicles in TCKO mice could be also due to the dysregulated RUNX2 function by loss of TETs proteins. We added some discussion in the revised manuscript in line 404-412.

2, Comments, No.6,"I did not think that bone formation in the bone marrow is accelerated, because it is unlikely. However, the previous data of Safranin O at P0 in TCKO mice showed many bone matrix than wild-type mice. The new Safranin O and fast green staining also showed abundant bone matrix. Therefore, Bone formation is likely to be not apparently impaired in TCKO mice in limbs. The transition of chondrocytes to osteoblasts is not impaired from the histological section at P0". This comment was concerned with bone formation and transition of chondrocytes to osteoblasts at P0. While the author's new data focused on E16.5 and/or E17.5, not P0. Thus, the author's reply to this comment was inappropriate and insufficient.

Thanks for the reviewer's suggestion. Our previous data focused on E16.5 and E17.5 and showed more Col10a1 and Mmp13 signals in the bone marrow of TCKO mice compared with WT mice (Supplementary Fig. 3b-3d). The reviewer concerned with bone formation and transition of chondrocytes to osteoblasts at P0. At P0, the 'bone matrix' of TCKO is a mixture of Safranin O and fast green staining (Fig. S3a and Figure R5a). We did COL2 immunofluorescence (Figure R5b) using the slides of P0 and found that TCKO mice displayed COL2 positive signal in 'bone matrix', implying that the transition of chondrocytes to osteoblasts at P0 were also delayed. We made some refinement for this point in the revised manuscript line 137-160.

Figure R5. **a** The enlarged picture of the Safranin O staining at P0 in Fig. S3a. The mixture of Safranin O and fast green staining was labeled by arrows. **b** The immunofluorescence of COL2 in the diaphysis of WT and TCKO bones at P0.

Reviewer #5 (Remarks to the Author):

In this paper, the authors suggest that TET enzymes function to regulate RUNX2 activity and play a role in a bone formation, especially in the differentiation of osteoblast both in calvarial bones (via intramembranous ossification) and femurs (via endochondral ossification). These results will be helpful to unravel molecular events occurring by TET enzyme in the bone development. The data are interesting and add to the literature. There are some shortcomings that could be addressed:

1. The authors used Prx1-Cre:Tet conditional knockout mice to show the impairment of bone formation and furthermore, they also showed that Tet interacts with RUNX2 to regulate osteogenesis. In limb, beside osteoprogenitors, chondroprogenitors are also most likely expressed Prx1. As well, RUNX2 is also expressed in chondrocyte and direct target genes of RUNX2 in chondrocytes are *Ihh*, *Col10a1*, and *Mmp13* in the limb. Moreover, it has been known that in the case of RUNX2 KO, chondrocyte-specific KO of RUNX2 induced severe dwarfism. However, bone-specific KO did not induce severe dwarfism compared to chondrocyte-specific KO of RUNX2. It is rather prevent the transition of cartilage to bone formation. Therefore, the process of the endochondral ossification (Supplementary Figure 3) more likely due to the delay in chondrocyte maturation (Chondrocyte hypertrophy) but not the delay in the transition of chondrocytes to osteoblasts (or osteoblast maturation).

Thanks a lot for the reviewer's comments and suggestions. The reviewer pointed out that *Prx1-Cre* could label both osteoprogenitors and chondroprogenitors. As TETs proteins could interact with RUNX2 and regulate the expression of RUNX2-related genes, the reviewer suggested that the impairment of the endochondral ossification (Supplementary Figure 3) is more likely due to the delay in chondrocyte maturation (Chondrocyte hypertrophy) but not the delay in the transition of chondrocytes to osteoblasts (or osteoblast maturation). We agree with the reviewer for this point in terms of the process of the endochondral ossification. We did in vitro chondrocyte differentiation assay and found that chondrocyte differentiation was also impaired in TCKO cells, compared with wild type chondrocytes (Supplementary Fig. 3k-3l). These data support that endochondral ossification is impaired due to the delay in chondrocyte maturation. We changed some of our conclusions and added discussion about the chondrocyte differentiation and maturation in the line 137-160 of revised manuscript.

2. In addition, since Prx1 interacts with RUNX2 as authors showed, there is high possibility that hypertrophic chondrocytes in the limb might be affected (Authors showed that direct target genes of RUNX2 in chondrocytes, *COI10a1* and *MMP13* had affected as well in the paper...pointing that it is rather chondrocyte maturation affected by Tet KO).

Since Prx1 and RUNX are expressed in both osteoblast and chondrocyte, the authors should interpret carefully the role of Prx1-Cre:Tet and RUNX2 during the process of endochondral ossification.

We thank the reviewer for the kind suggestions. Previously, we mainly focused on the regulation of RUNX2 target genes in osteoblasts by TET proteins. As the reviewer pointed that RUNX2 target genes, including *Col10a1* and *MMP13* in chondrocytes were also affected in TCKO mice (Supplementary Fig. 3b-3d), indicating that the DNA methylation also play key roles in the regulation of the genes critical for endochondral ossification. We analyzed the *Col10a1* and *MMP13* in the high-throughput sequencing data for some clues. Both *Col10a1* and *MMP13* were occupied by RUNX2 (Supplementary Fig. 10g-10f). Loss of TET proteins decreased the chromatin accessibility of *Col10a1* and *MMP13* with increased 5mC level and decreased 5hmC level (Supplementary Fig. 10g-10f). Although we did not use the primary chondrocytes for the study, these data indicated that *Col10a1* and *MMP13* might be also regulated in chondrocytes by the interaction of TET proteins and RUNX2. However, the detailed mechanism how TET proteins and RUNX2 collectively regulated the gene expression of *Col10a1* and *MMP13* is worthy to be investigated in further studies. We added this discussion in the revised manuscript line 390-403.

Supplementary Fig 10. (g-f) Screenshot of *Col10a1* and *MMP13* in CG methylation level from WGBS, 5hmC-Seal, ATAC-seq in WT and TCKO primary osteoblasts.

1. Omatsu, Y. *et al.* Runx1 and Runx2 inhibit fibrotic conversion of cellular niches for hematopoietic stem cells. *Nature Communications* **13**, 2654 (2022).
2. Liu, Z. *et al.* Mediator MED23 cooperates with RUNX2 to drive osteoblast differentiation and bone development. *Nat Commun* **7**, 11149 (2016).
3. Suo, J. *et al.* VGLL4 promotes osteoblast differentiation by antagonizing TEADs-inhibited Runx2 transcription. *Sci Adv* **6**(2020).

REVIEWERS' COMMENTS

Reviewer #4 (Remarks to the Author):

I have no more questions.